# Plasticity as the Mirror of Empowerment

**David Abel**
Google DeepMind

**Michael Bowling**
Amii, University of Alberta

**André Barreto**
Google DeepMind

**Will Dabney**
Google DeepMind

**Shi Dong**
Google DeepMind

**Steven Hansen**
Google DeepMind

**Anna Harutyunyan**
Google DeepMind

**Khimya Khetarpal**
Google DeepMind

**Clare Lyle**
Google DeepMind

**Razvan Pascanu**
Google DeepMind, Mila

**Georgios Piliouras**
Google DeepMind

**Doina Precup**
Google DeepMind

**Jonathan Richens**
Google DeepMind

**Mark Rowland**
Google DeepMind

**Tom Schaul**
Google DeepMind

**Satinder Singh**
Google DeepMind

## Abstract

Agents are minimally entities that are influenced by their past observations and act to influence future observations. This latter capacity is captured by empowerment, which has served as a vital framing concept across artificial intelligence and cognitive science. This former capacity, however, is equally foundational: In what ways, and to what extent, can an agent be influenced by what it observes? In this paper, we ground this concept in a universal agent-centric measure that we refer to as *plasticity*, and reveal a fundamental connection to empowerment. Following a set of desiderata on a suitable definition, we define plasticity using a new information-theoretic quantity we call the *generalized directed information*. We show that this new quantity strictly generalizes the directed information introduced by Massey (1990) while preserving all of its desirable properties. Under this definition, we find that plasticity is well thought of as the mirror of empowerment: The two concepts are defined using the same measure, with only the direction of influence reversed. Our main result establishes a tension between the plasticity and empowerment of an agent, suggesting that agent design needs to be mindful of both characteristics. We explore the implications of these findings, and suggest that plasticity, empowerment, and their relationship are essential to understanding agency.

## 1 Introduction

Two capacities represent fundamental aspects of any agent, regardless of its design, physical substrate, or goals: In what ways can the agent be shaped by what it observes? And, what things can an agent *do* to shape aspects of its observable environment? A system that entirely lacks either capacity—for instance, a machine whose outputs cannot be influenced by its inputs—is likely lacking agency altogether (Barandiaran et al., 2009). In this paper, we highlight a fundamental connection between these two capacities, which we refer to as *empowerment* and *plasticity*.

**Empowerment.** Empowerment refers to an agent's ability to influence its observable future. For example, a car with a full tank of gas has higher empowerment than a car with an empty tank, precisely because the former can realize more outcomes than the latter. The concept of empowerment was first introduced by Klyubin et al. (2005b) as "an agent-centric quantification of the amount of control or influence the agent has and perceives" (p. 2). Since its inception, the concept has been expanded

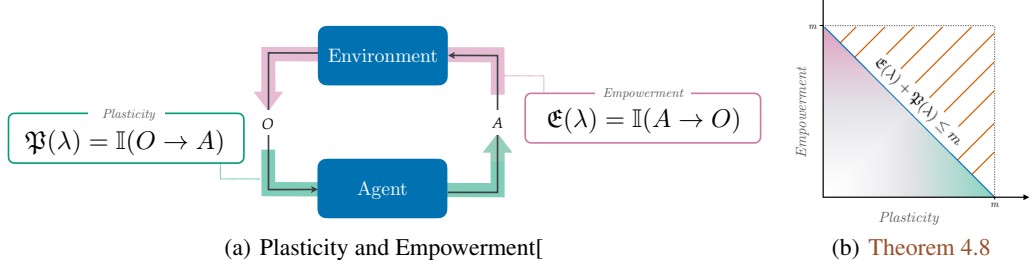

(a) Plasticity and Empowerment[

(b) Theorem 4.8

Figure 1: On the left, we depict the mirror of plasticity and empowerment. On the right, we show the tight upper-bound on an agent's plasticity $\mathfrak{P}(\lambda)$ and empowerment $\mathfrak{E}(\lambda)$ established by Theorem 4.8: While values of $\mathfrak{P}(\lambda)$ and $\mathfrak{E}(\lambda)$ from zero up to an interface- and interval-dependent constant $m$ are realizable, their sum can be no greater than $m$.

to act as a plausible explanation of intrinsic drive in the absence of reward (Klyubin et al., 2005a; Mohamed and Jimenez Rezende, 2015; Ringstrom, 2023), framed key aspects of safety and alignment (Salge and Polani, 2017; Turner et al., 2021), skill discovery (Gregor et al., 2016; Campos et al., 2020; Baumli et al., 2021; Levy et al., 2023), social influence (Jaques et al., 2019), self-preservation (Ringstrom, 2023), and has played a meaningful role in explanatory theories of how people explore (Brändle et al., 2023).

**Plasticity.** Conversely, plasticity reflects an agent's capacity to "remain...adaptive... in response to significant events" (p. 2, Carpenter and Grossberg 1988), and has been studied extensively across neuroscience, biology, and machine learning. Within neuroscience, "plasticity" often refers to *synaptic* plasticity, and reflects "the capacity of the neural activity generated by an experience to modify neural circuit function" (p. 1, Citri and Malenka, 2008). It is a central building block for understanding memory and forgetting (Fusi et al., 2005), often studied through the lens of the stability-plasticity dilemma (Carpenter and Grossberg, 1988, 1987a,b). In other areas of biology, plasticity is treated roughly as "the power of an organism to adapt action" (p. 531, Wheeler 1910), or more generally the "environmental responsiveness" of an organism (p. 34, West-Eberhard, 2003). Within machine learning, plasticity is often studied under the lens of the *loss* of plasticity (Dohare et al., 2021, 2024), which is especially prevalent within neural networks (Lyle et al., 2023; Abbas et al., 2023). Each of Chen et al. (2023), Raghavan and Balaprakash (2021), and Kumar et al. (2023) further study plasticity under the lens of the stability-plasticity dilemma in different learning settings. Across all of this work, it is clear plasticity is an important capacity for any agent—its connection to empowerment, however, beyond sometimes sharing an information-theoretic toolkit (Kumar et al., 2023), has remained distant. To this end, we develop a simple, universal definition of agent plasticity (Definition 4.1) that brings the concept on equal footing with empowerment.

**The Mirror of Plasticity and Empowerment.** Our first finding is that *plasticity is well thought of as the mirror of empowerment* (Proposition 4.6), pictured in Figure 1(a). To reveal this mirror, we exploit the inherent symmetry in the exchange of symbols between agent and environment, as in the bidirectional communication model introduced by Marko (1973). This allows us to consider both the agent and environment's point of view: How much power does the environment have to shape the signals that the agent emits, and how malleable is the agent to be shaped by the environment? These two quantities constitute the agent's plasticity and the environment's empowerment, and Proposition 4.6 shows they are identical. In other words, the concepts of plasticity and empowerment are well-defined using the same measure, with only the direction of influence reversed. To build toward this framing, we develop the *generalized directed information* (GDI, Definition 3.1) as a suitable formalism for the concept of plasticity under minimal assumptions. The GDI is a strict generalization of the directed information (Proposition 3.2) introduced by Massey (1990). Through a new extension of the conservation law of information (Theorem 3.5), we show that all of the information exchanged between any agent and environment can be broken into either plasticity or empowerment. This conservation unlocks our main result (Theorem 4.8), which proves that empowerment and plasticity can be in tension as pictured in Figure 1(b). Consequently, the design of an agent implicitly faces a dilemma with respect to plasticity and empowerment. We take this

result to have significant implications for the design and analysis of agents, and suggest that further examining this dilemma is an important direction for future research. We further highlight the necessary and sufficient conditions for an agent to admit non-zero plasticity (Lemma 4.2), and exploit these conditions to prove several intuitive properties hold of plasticity (Theorem 4.3). Collectively, we take these findings to provide a rich toolkit with which to study agents under minimal assumptions.

## 2 Preliminaries: Directed Information, Agents, and Empowerment

**Notation.** Throughout, we let calligraphic capital letters $\mathcal{X}, \mathcal{Y}$ denote sets, and capital italics $X, Y$ denote discrete random variables over their respective domains $\mathcal{X}$ and $\mathcal{Y}$. We let $X_{1:n} = (X_1, \ldots, X_n)$ express a sequence of $n$ discrete random variables, and let $[a : b]_n$ denote an inclusive interval of integers such that $1 \leq a \leq b \leq n$. Lastly, we let $\Delta(\mathcal{X})$ denote the probability simplex over the set $\mathcal{X}$. That is, the function $p : \mathcal{X} \to \Delta(\mathcal{Y})$ expresses a probability mass function $p(\cdot \mid x)$, over $\mathcal{Y}$, for each $x \in \mathcal{X}$. All notation and core definitions are summarized in Table 1 in Appendix A.

### 2.1 Directed Information

We assume the basic definitions of information theory (Shannon, 1948) are known to the reader, such as the Shannon entropy $\mathbb{H}(X)$ of a discrete random variable $X$ supported on the set $\mathcal{X}$, and the mutual information $\mathbb{I}(X; Y)$ between two discrete random variables $X$ and $Y$ supported on $\mathcal{X}$ and $\mathcal{Y}$ respectively. We will additionally make heavy use of the *directed information*, developed by Massey (1990) and expanded on by Kramer (1998), and Massey and Massey (2005), defined as follows.

**Definition 2.1.** *For any two sequences of discrete random variables, $X_{1:n} = (X_1, \ldots X_n)$ and $Y_{1:n} = (Y_1, \ldots, Y_n)$ the **directed information** is*

$$\mathbb{I}(X_{1:n} \to Y_{1:n}) \stackrel{\text{def}}{=} \sum_{i=1}^{n} \mathbb{I}(X_{1:i}; Y_i \mid Y_{1:i-1}). \tag{1}$$

The directed information is designed to capture the influence that the past elements $X_i$ have on future elements $Y_i$. This notation assumes each $X_i$ precedes and is a possible cause of $Y_i$, though if we want to consider indexing where $Y_i$ precedes $X_i$ we make use of the $\hookrightarrow$ in our notation,

$$\mathbb{I}(Y_{1:n} \hookrightarrow X_{1:n}) \stackrel{\text{def}}{=} \sum_{i=2}^{n} \mathbb{I}(Y_{1:i-1}; X_i \mid X_{1:i-1}). \tag{2}$$

Directed information is particularly useful for analyzing systems with *feedback*—as is clearly present in the case of the information exchange between agent and environment in reinforcement learning. In fact, directed information was originally proposed in the context of the bidirectional communication theory developed by Marko (1973). We recall two facts about the directed information that we will later exploit and generalize. First, directed information is non-negative and upper-bounded.

**Proposition 2.2** (Adapted from Theorem 2 of Massey, 1990 and Property 3.2 of Kramer, 1998)**.** *For any two sequences of discrete random variables $X_{1:n} = (X_1, \ldots, X_n)$, $Y_{1:n} = (Y_1, \ldots, Y_n)$,*

$$0 \leq \mathbb{I}(X_{1:n} \to Y_{1:n}) \leq \mathbb{I}(X_{1:n}; Y_{1:n}), \tag{3}$$

*where equality holds in the left inequality if and only if $\mathbb{I}(X_{1:i}; Y_i \mid Y_{1:i-1}) = 0$ and in the right inequality if and only if $\mathbb{H}(Y_i \mid Y_{1:i-1}, X_{1:i}) = \mathbb{H}(Y_i \mid Y_{1:i-1}, X_{1:n})$, for all $i = 1, 2, \ldots, n$.*

The conditions for equality on the upper bound indicate that the directed information is equal to the mutual information when there is only communication flowing in one direction. We next confirm this intuition by recalling one of the central facts about directed information proven by Massey and Massey (2005)—that directed information is *conserved* in a particular sense.

**Proposition 2.3** (Conservation Law of Directed Information; adapted from Proposition 2 of Massey and Massey, 2005)**.** *For any pair $(X_{1:n}, Y_{1:n})$,*

$$\mathbb{I}(X_{1:n}; Y_{1:n}) = \mathbb{I}(X_{1:n} \to Y_{1:n}) + \mathbb{I}(Y_{1:n} \hookrightarrow X_{1:n}). \tag{4}$$

*where $\mathbb{I}(Y_{1:n} \hookrightarrow X_{1:n}) = \sum_{i=2}^{n} \mathbb{I}(Y_{1:i-1}; X_i \mid X_{1:i-1})$.*

That is, the amount that $X_{1:n}$ influences $Y_{1:n}$ together with the amount that $Y_{1:n}$ influences $X_{1:n}$ is exactly the total mutual information between $X_{1:n}$ and $Y_{1:n}$. This highlights why the presence of bidirectional feedback forces $\mathbb{I}(X_{1:n} \to Y_{1:n})$ to be strictly less than $\mathbb{I}(X_{1:n}; Y_{1:n})$—notice that if $\mathbb{I}(Y_{1:n} \hookrightarrow X_{1:n}) = 0$, then we arrive at the identity $\mathbb{I}(X_{1:n}; Y_{1:n}) = \mathbb{I}(X_{1:n} \to Y_{1:n})$.

## 2.2 Agents

Alongside the tools of information theory, the primary objects of our study are *agents* and the *environments* they interact with. Inspired by the work on general RL (Hutter, 2004; Hutter et al., 2024) and following the recent trend to move beyond the Markov property (Dong et al., 2022; Lu et al., 2023; Bowling et al., 2023; Abel et al., 2023a, 2024; Bowling and Elelimy, 2025), we embrace a perspective on the agent-environment interaction that is intentionally light on assumptions. Our starting point is to define a pair of finite sets that characterize the possible signals exchanged between agent and environment (Dong et al., 2022).

**Definition 2.4.** *The **interface** is a pair of finite sets,* $(\mathcal{A}, \mathcal{O})$ *such that* $|\mathcal{A}| \geq 2, |\mathcal{O}| \geq 2$.

We refer to each element $a \in \mathcal{A}$ as an action, and each element $o \in \mathcal{O}$ as an observation. Together, these two sets form the space of possible interactions between any agent and environment that share an interface. Following the conventions of Hutter (2000); Dong et al. (2022); Abel et al. (2023a) and Bowling et al. (2023), we refer to this set of possible interactions of any length as *histories*. To accommodate the case where the environment emits the first symbol *or* the agent emits the first symbol, we distinguish between sequences that end with an action and end with an observation.

**Definition 2.5.** *The **histories**, **histories ending in action**, and **histories ending in observation**, that arise from interface* $(\mathcal{A}, \mathcal{O})$ *are defined as the sets*

$$\mathcal{H} \overset{\text{def}}{=} \mathcal{H}_{..\mathcal{A}} \cup \mathcal{H}_{..\mathcal{O}}, \tag{5}$$

$$\mathcal{H}_{..\mathcal{A}} \overset{\text{def}}{=} (\mathcal{O} \times \mathcal{A})^* \cup (\mathcal{A} \times \mathcal{O})^* \times \mathcal{A}, \tag{6}$$

$$\mathcal{H}_{..\mathcal{O}} \overset{\text{def}}{=} (\mathcal{A} \times \mathcal{O})^* \cup (\mathcal{O} \times \mathcal{A})^* \times \mathcal{O}. \tag{7}$$

We refer to any element $h \in \mathcal{H}$ as a history.

An agent, then, is a stochastic mapping from a history to an action, as in the agent functions introduced by Russell and Subramanian (1994).

**Definition 2.6.** *An **agent** relative to interface* $(\mathcal{A}, \mathcal{O})$ *is a function,* $\lambda : \mathcal{H}_{..\mathcal{O}} \to \Delta(\mathcal{A})$.

Lastly, environments are stochastic functions that produce observations for an agent.

**Definition 2.7.** *An **environment** relative to interface* $(\mathcal{A}, \mathcal{O})$ *is a function,* $e : \mathcal{H}_{..\mathcal{A}} \to \Delta(\mathcal{O})$.

We let $\Lambda_{\text{all}}$ and $\mathcal{E}_{\text{all}}$ denote the set of all such functions $\lambda$ and $e$ defined over interface $(\mathcal{A}, \mathcal{O})$ respectively, and let $\Lambda \subseteq \Lambda_{\text{all}}$ and $\mathcal{E} \subseteq \mathcal{E}_{\text{all}}$ denote non-empty subsets of $\Lambda_{\text{all}}$ or $\mathcal{E}_{\text{all}}$.

Together, an agent $\lambda$ and environment $e$ that share an interface interact by exchanging observation and action signals, inducing the stochastic process $A_1 O_1, A_2 O_2 \ldots A_n O_n \ldots$ that may carry on indefinitely. We assume that actions are emitted first, but this is purely for convenience and the decision is ultimately benign. Under this framing, it is useful to call attention to a symmetry between agent and environment.

**Remark 2.8.** *Agents and environments, as defined, are symmetric in the sense that swapping sets* $\mathcal{A}, \mathcal{O}$ *converts every agent into an environment, and every environment into an agent. To see this, notice that an environment,* $e : \mathcal{H}_{..\mathcal{A}} \to \Delta(\mathcal{O})$ *becomes a function from* $\mathcal{H}_{..\mathcal{O}}$ *to* $\Delta(\mathcal{A})$, *which is precisely the definition of an agent.*

## 2.3 Empowerment

Empowerment was first introduced by Klyubin et al. (2005b) as "how much control or influence an [agent] has" (p. 2). More precisely, Klyubin et al. focus on the *open-loop agents* represented by a probability mass function $p(a^n) \in \Lambda_{a^n}$ over the sequence of $n$ discrete random variables, $A_{1:n} = (A_1, A_2, \ldots, A_n)$. Then, the empowerment of an agent is the maximum mutual information between this sequence of actions, $A_{1:n}$, and the observation produced directly following the sequence, $O_n$, as follows.

$$\mathfrak{E}_{\text{K}}^n(\Lambda_{a^n}, e) = \max_{p(a^n) \in \Lambda_{a^n}} \mathbb{I}(A_{1:n}; O_n). \tag{8}$$

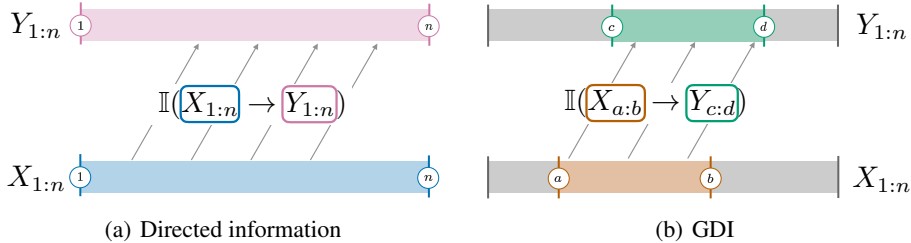

(a) Directed information        (b) GDI

Figure 2: A visual of the directed information (left) and generalized directed information (right). In the case of GDI, the intervals can fully or partially overlap, or not overlap at all. When the intervals are identical, the two quantities are identical (Proposition 3.2).

**Empowerment as Directed Information.** Capdepuy (2011) build on Klyubin's definition of empowerment by allowing the agent and the environment to *both* communicate with one another. Thus, under Capdepuy's definition, the agents in question can be closed-loop, which modifies the definition of empowerment in two ways. First, the max is instead taken with respect to a richer class of functions such as $\Lambda_{\text{all}}$, compared to $\Lambda_{a^n}$. Second, the mutual information is replaced by the direction information, as follows,

$$\mathfrak{E}_{\text{C}}^n(\Lambda, e) = \max_{\lambda \in \Lambda} \mathbb{I}(A_{1:n} \to O_{1:n}). \tag{9}$$

In light of its generality, we take the directed information view of empowerment to be the appropriate starting point.

## 3 Generalized Directed Information

A significant limitation of directed information is that it requires both sequences $X_{1:n}$ and $Y_{1:n}$ to be of the same length, and start from the beginning of time. Thus, the existing measures do not allow us to measure the empowerment from one sequence of arbitrary length to another sequence of arbitrary length. To remedy this, we here develop a new measure that extends the directed information to arbitrary sequences $X_{a:b}$ and $Y_{c:d}$ that captures directed information as a special case (Proposition 3.2). We define this new measure as follows.

**Definition 3.1.** *Let $X_{1:n}, Y_{1:n}$ be sequences of discrete random variables and let $[a:b]_n$ and $[c:d]_n$ be valid intervals. The **generalized directed information** (GDI) is defined as:*

$$\mathbb{I}(X_{a:b} \to Y_{c:d}) \stackrel{\text{def}}{=} \sum_{i=\max(a,c)}^{d} \mathbb{I}(X_{a:\min(b,i)}; Y_i \mid X_{1:a-1}, Y_{1:i-1}). \tag{10}$$

As with directed information, this notation assumes $X_i$ precedes and is a possible cause of $Y_i$. If we want to consider the indexing such that $Y_i$ precedes $X_i$ we again use $\hookrightarrow$ in our notation (Equation 2), $\mathbb{I}(X_{a:b} \hookrightarrow Y_{c:d}) \stackrel{\text{def}}{=} \sum_{i=\max(a+1,c)}^{d} \mathbb{I}(X_{a:\min(b,i-1)}; Y_i \mid X_{1:a-1}, Y_{1:i-1})$.

To ensure we capture the spirit of the original directed information, we verify that several properties hold of our generalized definition. Namely, that (1) when the sequences $X_{a:b}$ and $Y_{c:d}$ are the same length and both begin at the start of time, we recover directed information, and (2) the basic properties of directed information such as the conservation law are maintained. We first note that when $a = c = 1$ and $b = d = n$, we recover the directed information, as desired.

**Proposition 3.2** (GDI Strictly Generalizes DI). *Let $a = c = 1$ and $b = d = n$. Then,*

$$\mathbb{I}(X_{a:b} \to Y_{c:d}) = \mathbb{I}(X_{1:n} \to Y_{1:n}). \tag{11}$$

Due to space constraints, we defer all proofs to Appendix C. Next, we show that if the first sequence comes *after* the second sequence, then the GDI is zero.

**Proposition 3.3** (Temporal Consistency of GDI). *If $a > d$ then $\mathbb{I}(X_{a:b} \to Y_{c:d}) = 0$.*

Thus, the quantity is only non-zero when at least some of the left-hand sequence of random variables precedes at least some of the right-hand sequence of random variables, as we would expect.

We next show that we can subdivide the intervals along which we measure the GDI.

**Proposition 3.4** (Summation of Intervals)**.** *Let* $a \leq k < b$ *and* $c \leq \ell < d$.

$$\mathbb{I}(X_{a:b} \to Y_{c:d}) = \mathbb{I}(X_{a:b} \to Y_{c:\ell}) + \mathbb{I}(X_{a:b} \to Y_{\ell+1:d}) \tag{12}$$
$$= \mathbb{I}(X_{a:k} \to Y_{c:d}) + \mathbb{I}(X_{k+1:b} \to Y_{c:d}). \tag{13}$$

Since each term on the right is also just a GDI term, we can thus decompose any pair of sequences of random variables into arbitrarily many valid sub-sequences, and preserve GDI.

We now prove the most significant property of the new measure: The GDI exhibits a conservation law similar to that of the directed information (Proposition 2.3). This conservation is especially important for our discussion of plasticity and empowerment, as it is critical for understanding their tension.

**Theorem 3.5** (Conservation Law of GDI)**.**

$$\mathbb{I}(X_{a:b}; Y_{c:d} \mid X_{1:a-1}, Y_{1:c-1}) = \mathbb{I}(X_{a:b} \to Y_{c:d}) + \mathbb{I}(Y_{c:d} \hookrightarrow X_{a:b}). \tag{14}$$

Note that the sum of the two GDI terms is a *conditional* mutual information term. There are two perspectives that help to understand this conditioning. First, notice that when $a = 1$ and $c = 1$, the terms $X_{1:0}$ and $Y_{1:0}$ disappear, and we recover the same term as in the original conservation law (Proposition 2.3). Second, conditioning on past information allows us to remove possible confounders. For example, imagine $X_1 = 0$, and this emission ensures that both $X_3 = 0$ and $Y_4 = 0$. Now, if we measure $\mathbb{I}(X_{2:4} \to Y_{3:5})$, the total information flowing from $X_3$ to $Y_4$ needs to remove the potential information from $X_1$—this is the reason for the conditioning.

We prove several other properties of the GDI including an extension of the data-processing inequality (Derpich and Østergaard, 2021), but defer these results to Appendix D due to space constraints.

## 4 Plasticity and Empowerment as Generalized Directed Information

We now motivate the GDI as a definition for plasticity. We start by enumerating a set of desiderata that characterize an account of what a suitable definition of plasticity might look like.

Our desiderata come in two forms. The first set offer a *structural* picture of the definition, while the second set focus on the *semantics* of the definition. While we hope that these desiderata are illuminating, we also note that they fall short of a complete axiomatic characterization of plasticity. In this sense, we anticipate that subtle aspects of the definition may need slight adjustment as we hone in on the right axioms for the concept (Lakatos, 1963). We suggest that identifying such a bedrock is an important direction for further research, but is here out of scope.

**Structural Desiderata.** A definition of agent plasticity should adhere to the following four structural criteria. First, the definition should be mathematically precise. That is, we should be able to codify all relevant commitments about our definition into a simple set of mathematical terms. Second, the definition should be simple to state, and only require minimal conceptual overhead in terms of new formalism. Third, the definition should not impose additional assumptions on our agent and environment setup beyond those already enumerated: Agent and environment share an interface consisting of two finite sets and interact in discrete time by exchanging signals from their interface. Fourth, the definition should provide flexibility to ground discussions of plasticity under a variety of settings. That is, the plasticity measure should be consistent with respect to arbitrary windows of time, or with or without a specific environment present. It is critical that we can measure the plasticity of an agent throughout its learning process, and not just the interval $[1:n]$. This fourth structural desiderata is what motivates the development of the GDI over regular directed information, as without it we cannot speak about plasticity over arbitrary intervals of experience.

**Semantic Desiderata.** Next, we identify desiderata on our definition that indicate what the definition should *mean*. These are inspired by a mixture of reflections from the broader communities that have studied plasticity, and a handful of thought experiments that stretch intuitions about plasticity. First, plasticity is always a non-negative scalar value. Second, agents that are unaffected by what they

observe have zero plasticity. For instance, a constant agent that always outputs the same action should have zero plasticity. Lastly, agents whose behavior is influenced by past observations have non-zero plasticity that is proportional to this influence. Notably, this places the emphasis of plasticity on the extent to which the agent can adapt its *behavior* rather than *cognitive* faculties of the agent, such as memories or other internal parameters. We return to this point in the discussion.

## 4.1 Plasticity

These desiderata motivate the GDI as our definition of plasticity. Intuitively, the definition captures *how much a sequence of observations influences a sequence of the agent's actions*. The GDI allows us to select which observation sequence $O_{a:b}$ and action sequence $A_{c:d}$ are of interest, as follows.

**Definition 4.1.** *The **plasticity** of agent $\lambda$ relative to $\mathcal{E}$ from $[a:b]_n$ to $[c:d]_n$ is defined as*

$$\mathfrak{P}_{\substack{a:b \\ c:d}}(\lambda, \mathcal{E}) \overset{\text{def}}{=} \max_{e \in \mathcal{E}} \mathbb{I}(O_{a:b} \to A_{c:d}). \tag{15}$$

For brevity, we will most often refer to an agent's plasticity relative to a set of environments as $\mathfrak{P}(\lambda, \mathcal{E})$ and obscure dependence on the interval when otherwise clear from context. Additionally, while we introduce the definition in its general form, we most often consider the plasticity of an agent interacting with a specific environment, $e$, which is recovered as a special case involving a singleton set, $\mathfrak{P}(\lambda, \{e\})$. We refer to such cases as $\mathfrak{P}(\lambda)$ or $\mathfrak{P}(\lambda, e)$ for brevity. This is similar to the distinction pointed out by Capdepuy (2011) on actual versus potential information flow: "It is important to understand that empowerment, because it is defined as a capacity, is a potential quantity. The actual controller of the agent is not considered." (p. 44). We embrace this distinction for plasticity as well, and modulate the set $\mathcal{E}$ to refer to either instantiated or potential plasticity. Lastly, we note that the above definition uses the convention where, for every $i = 1, \ldots, n$, each $A_i$ precedes each $O_i$, but where needed, we can also make use of the $\hookrightarrow$ notation.

**GDI and the Desiderata.** We next confirm that the GDI satisfies our presented structural and semantic desiderata. For the first two structural desiderata, we call attention to the fact that the quantity $\mathfrak{P}(\lambda, \mathcal{E})$ is mathematically precise, and requires minimal overhead (albeit subjectively): While the GDI *is* new, it provides a necessary enrichment of the directed information to account for (1) sequences of arbitrary length, (2) that may not start at the beginning of time, and (3) involve systems with bidirectional communication. These properties are critical to satisfy our remaining structural desiderata, and thus we believe this slight added complexity is earned. Third, our definition imposes no further assumptions on agent or environment. We inherit the general setting of an agent interacting with an environment in discrete time, and add no further restrictions. Lastly, as noted, our fourth structural desiderata is accounted for in virtue of the GDI: We are now able to range over arbitrary sequences, and can accommodate the cases when a specific environment $e$, a set $\mathcal{E}$, or the set of all environments $\mathcal{E}_{\text{all}}$ are under consideration. For the semantic desiderata, we summarize the fact that $\mathfrak{P}$ adheres to these desiderata using the following two results. The first is a lemma that isolates the necessary and sufficient conditions for plasticity relative to a specific environment to be non-zero.

**Lemma 4.2** (Necessary and Sufficient Conditions for Positive Plasticity)**.** *An agent has positive plasticity relative to $e$ if and only if there is a time-step where actions are influenced by observation.*

*More formally, for any pair $(\lambda, e)$ and indices $[a:b]_n$, $[c:d]_n$, there exists an $i \in [\max(a,c):d]$ such that*

$$\mathbb{H}(A_i \mid O_{1:a-1}, A_{1:i-1}) > \mathbb{H}(A_i \mid O_{a:\min(b,i)}, O_{1:a-1}, A_{1:i-1}) \tag{16}$$

*if and only if $\mathfrak{P}_{\substack{a:b \\ c:d}}(\lambda, e) > 0$.*

Notice that the only difference between the left and right side of the inequality is that the right entropy term also conditions on $O_{a:\min(b,i)}$—consequently, the inequality holds when learning about the observations from $[a:\min(b,i)]$ *lowers* uncertainty about $A_i$. As a result, a pair $(\lambda, e)$ will have non-zero plasticity from interval $[a:b]_n$ to $[c:d]_n$ if and only if the observations in $[a:b]_n$ provides information about the actions in $[c:d]_n$. We make heavy use of this lemma to elucidate the following general properties of plasticity.

**Theorem 4.3.** *The following properties hold of the function $\mathfrak{P}$, for all intervals $[a:b]_n, [c:d]_n$:*

*(i) For all $(\lambda, \mathcal{E})$, $\mathfrak{P}_{\substack{a:b \\ c:d}}(\lambda, \mathcal{E}) \geq 0$.*

*(ii) For any $a < d$, there exists a pair $(\lambda, \mathcal{E})$ where $\lambda$ is deterministic and $\mathfrak{P}_{c:d}^{a:b}(\lambda, \mathcal{E}) > 0$.*

*(iii) For all $\mathcal{E}_{\text{small}} \subseteq \mathcal{E}_{\text{big}}$, $\mathfrak{P}_{c:d}^{a:b}(\lambda, \mathcal{E}_{\text{small}}) \leq \mathfrak{P}_{c:d}^{a:b}(\lambda, \mathcal{E}_{\text{big}})$.*

*(iv) (Informal) The following agents all have zero plasticity relative to every environment set: Open-loop agents, agents that always output the same action distribution, agents whose actions only depend on history length, and agents whose actions only depend on past actions.*

Together, this theorem and lemma indicate that the semantic desiderata are satisfied from a number of perspectives. By the necessary and sufficient conditions of Lemma 4.2, we find an intuitive test for whether an agent has plasticity in a given window: Are its actions impacted by its observations? The *magnitude* of this impact directly determines the magnitude of the agent's plasticity, aligned with the fourth desiderata. The lemma also indicates that many popular learning algorithms such as UCB1 (Auer et al., 2002) or Q-learning (Watkins and Dayan, 1992) have non-zero plasticity in most environments, as expected. Theorem 4.3 illustrates basic properties of plasticity: Point (i) confirms that plasticity is a non-negative scalar, while point (ii) shows that a deterministic agent can be plastic. Point (iii) highlights a form of monotonicity to plasticity—as the environment set under consideration is enriched, an agent's plasticity cannot go down. Lastly, property (iv) shows that many trivial agents have zero plasticity, such as agents whose actions are determined solely by factors *other* than what they observe such as open-loop or constant agents.

## 4.2 Revisiting Empowerment through GDI

In addition to capturing plasticity, the GDI can enrich our definition of empowerment as well. As discussed in our exposition of empowerment, the definition of Capdepuy (Equation 9) moves from the mutual information to the directed information. Due to the generality of the GDI, empowerment can further benefit from the flexibility of the GDI as follows.

**Definition 4.4.** *The **empowerment** of $\Lambda$ in environment $e$ from $[a:b]_n$ to $[c:d]_n$ is defined as*

$$\mathfrak{E}_{c:d}^{a:b}(\Lambda, e) \overset{\text{def}}{=} \max_{\lambda \in \Lambda} \mathbb{I}(A_{a:b} \to O_{c:d}). \tag{17}$$

As we might hope, when $1 = a = c$ and $b = d = n$, we recover the definition from Capdepuy, and when we restrict to the set of open loop agents $\Lambda_{a^n}$, we recover the definition from Klyubin et al.

**Proposition 4.5.** $\mathfrak{E}_{1:n}^{1:n}(\Lambda, e) = \mathfrak{E}_{\text{C}}^{n}(\Lambda, e)$ *and* $\mathfrak{E}_{n:n}^{1:n}(\Lambda_{a^n}, e) = \mathfrak{E}_{\text{K}}^{n}(\Lambda_{a^n}, e)$.

Thus, we can specialize to either past definition as desired. However, empowerment now benefits from the flexibility of the GDI to refer to an agent's empowerment over arbitrary windows of experience that need to begin at the start of time. Furthermore, we can recover a meaningful notion of the empowerment of a *single agent* by considering a singleton set, $\mathfrak{E}(\{\lambda\}, e)$, as with plasticity. We will again refer to this simply by the notation $\mathfrak{E}(\lambda)$ or $\mathfrak{E}(\lambda, e)$ for brevity when clear from context.

## 4.3 Main Result: Plasticity and Empowerment Tension

We next analyze the relationship between plasticity and empowerment. First, we highlight the fact that plasticity and empowerment are really two sides of the same coin. In one sense, this is obvious, as they are both defined as the GDI between action and observation with only the direction of influence reversed. There is a further sense, however, that connects them: The environment and the agent, as defined, are symmetric objects, since they are each a stochastic function from a history to a symbol. The names of the symbols and the functions are so far arbitrary. As a result, we can also ask about the *empowerment of the environment*, or the *plasticity of the environment* simply by taking the environment-centric view point suggested by Remark 2.8.

**Proposition 4.6** (Plasticity is the Mirror of Empowerment). *Given any pair $(\lambda, e)$ that share an interface, and valid intervals $[a:b]_n, [c:d]_n$, the agent's empowerment is equal to the plasticity of the environment, and the agent's plasticity is equal to the empowerment of the environment:*

$$\mathfrak{E}_{c:d}^{a:b}(\lambda) = \mathfrak{P}_{c:d}^{a:b}(e), \qquad \mathfrak{P}_{c:d}^{a:b}(\lambda) = \mathfrak{E}_{c:d}^{a:b}(e). \tag{18}$$

As a corollary, we find that despite our best efforts to design an agent with high plasticity or empowerment, some environments force *all* agents they interact with to have zero of either quantity.

**Corollary 4.7.** *For all* $(\lambda, e)$, $\mathfrak{P}(e) = 0 \iff \mathfrak{E}(\lambda) = 0$ *and* $\mathfrak{E}(e) = 0 \iff \mathfrak{P}(\lambda) = 0$.

In other words, even the most pliable or powerful agents cannot do anything meaningful in specific environments; the environment can always just choose to ignore the agent's actions, which places a firm limit on how empowered an agent can be. Similarly, even if we design a highly plastic agent, the agent can only make use of that plasticity in an environment with information.

**Example: Two-Player Game of Poker.** To build intuition, consider the case where the environment is itself another agent. That is, two agents—Alice and Bob—interact by exchanging actions. The interface is defined by two sets, $\mathcal{A}_{\text{alice}}, \mathcal{A}_{\text{bob}}$, where the actions of Alice are Bob's observations, and vice-versa. In this case, over any choice of intervals, *Alice's plasticity is identical to Bob's empowerment* and vice-versa. Imagine further that Alice and Bob are playing poker against each other, with some initial pool of chips. If either agent were to run out of chips, they are removed from play. Notice that the set of available actions to each agent is a strictly increasing function of their available pool of chips, as the larger pile of chips yields more bets that can be placed. Therefore, the more chips Alice or Bob has, the more empowered they are. And, by contrast, if Alice has more chips, this raises how plastic Bob can be, as he has a higher potential to be influenced by Alice's actions.

**Main Result.** We next present our main result, which highlights a fundamental tension between plasticity and empowerment. That is, if we consider a specific agent $\lambda$ interacting with an environment $e$, an important question arises: Is there any connection between the agent's level of plasticity, and the agent's level of empowerment? We answer this question by establishing a tight upper bound on the sum of the two quantities, which establishes a tension between an agent's plasticity and its empowerment over the same interval.

---

**Theorem 4.8** (Plasticity-Empowerment Tension). *For all intervals* $[a:b]_n$, $[c:d]_n$ *and any pair* $(\lambda, e)$, *let* $m = \min\{(b - a + 1) \log |\mathcal{O}|, (d - c + 1) \log |\mathcal{A}|\}$. *Then,*

$$\mathfrak{E}_{\substack{a:b \\ c:d}}(\lambda, e) + \mathfrak{P}_{\substack{c:d \\ a:b}}(\lambda, e) \leq m, \tag{19}$$

*is a tight upper bound on empowerment and plasticity. Furthermore, under mild assumptions on the interface and interval, there exists a pair* $(\lambda^\diamond, e^\diamond)$ *such that* $\mathfrak{P}_{\substack{c:d \\ a:b}}(\lambda^\diamond, e^\diamond) = m$, *and there exists a pair* $(\lambda', e')$ *such that* $\mathfrak{E}_{\substack{a:b \\ c:d}}(\lambda', e') = m$, *thereby forcing the other quantity to be zero.*

---

Therefore, an agent cannot simultaneously maximize its plasticity *and* its empowerment in a given pair of intervals, $[a:b]_n$, $[c:d]_n$. In fact, the achievable levels of empowerment for any pair of intervals are strictly determined by the agent's plasticity in those intervals (and vice versa), as pictured in Figure 1(b). The latter part of the result indicates that in extreme cases, a high plasticity can ensure that the agent has *zero* empowerment, and vice versa. Notably, the tension only comes about when we consider $\mathfrak{P}_{\substack{c:d \\ a:b}}$ and $\mathfrak{E}_{\substack{a:b \\ c:d}}$ over the same pair of intervals, $[a:b]_n$, $[c:d]_n$. Therefore, this tension may dissipate as the intervals change, or if one interval fully comes after the other; in this way, agents may oscillate between phases of high plasticity and high empowerment, but cannot simultaneously maximize both. A deeper examination of this trade-off and its implications for the design of agents is a significant opportunity for future research.

## 5 Discussion

In this paper, we have presented a careful study of plasticity and empowerment, and established a connection between them. Our first technical contribution is the development of the generalized directed information (GDI; Definition 3.1), which we take to be of independent interest across the many fields that make use of information theory. We identify several fundamental properties of the GDI, including a new conservation law (Theorem 3.5). We then motivate the GDI as an appropriate definition for the concept of plasticity, and provide a general definition of agent plasticity. We then identify the necessary and sufficient conditions for an agent to have non-zero plasticity (Lemma 4.2). Our analysis further reveals that plasticity admits a number of intuitive properties (Theorem 4.3), and that the GDI generalizes existing notions of empowerment (Proposition 4.5). We then discover a general new trade-off between plasticity and empowerment (Theorem 4.8), suggesting that we need to thoughtfully engage with both concepts in the study and design of agents. We take these results to serve as a meaningful bedrock from which we can study agency under minimal assumptions.

**Plasticity via Action, Policy, or Agent State?**    As discussed, the original conception of plasticity from the neuroscience literature is terms of *synaptic* plasticity—that is, the degree of flexibility inherent to the neural circuitry of a biological organism. By contrast, our account of plasticity focuses purely on *behavior*: We measure the influence of observation on *action*. This is more in line with how the biology community treats plasticity, as discussed in Chapter 30 of the book by Wheeler (1910). Our plasticity definition can be thought of as a special case of a broader family of definitions that consider the degree that observations influence various aspects of an agent, such as the agent's policy, action, or internal state. We can choose from many such options, and our formalism (and the corresponding theoretical results) will largely look the same. Consider the case of policies. Here, we suppose that every agent maintains a policy that belongs to a finite class of representable policies, $\Lambda_{\mathrm{B}} \subset \Lambda_{\mathrm{all}}$ (the "agent basis" from Abel et al., 2023a). Then, letting $L_1, \ldots L_n$ denote a sequence of discrete random variables over $\Lambda_{\mathrm{B}}$, we can express a policy-focused notion of plasticity in terms of the quantity $\mathbb{I}(O_{a:b} \to L_{c:d})$. This measures the degree to which observations influence which policy an agent chooses. With some added detail obscured, this is a strict generalization of Definition 4.1, but we choose to center around the action variant due to its simplicity. Alternatively, we can shift the influence of interest to *agent state* to better capture how well an agent can adapt its internal circuity in light of experience. In this case, we model each agent as maintaining an agent state (Dong et al., 2022; Abel et al., 2023b; Kumar et al., 2023). Letting $S_1, \ldots, S_n$ denote the sequence of random variables over this agent state space $\mathcal{S}$, we can recover our conception of plasticity by building around $\mathbb{I}(O_{a:b} \to S_{c:d})$. We believe this view is especially promising to examine agency under resource constraints (Simon, 1955; Russell and Subramanian, 1994), the extended mind (Clark and Chalmers, 1998; Harutyunyan, 2020), and the stability-plasticity dilemma (Carpenter and Grossberg, 1988), but leave exploration of these directions for future work.

**Plasticity, Empowerment, and Goals.**    Our framing is absent of the discussion of goals, despite their central role to agency. Indeed, as Ball states, "An agent does more than just alter its environment...they alter with intent" (p. 9, Ball 2023). We have here concentrated entirely on two basic influences; plasticity as the influence of environment on agent, and empowerment as the influence of agent on environment. It is important to enrich this story with some account of goals. We foresee several pathways for our theory to accommodate goals. For instance, we can treat reward as a special element of each observation. Then, we might isolate a special kind of influence that amounts to the agent being influenced by rewards (and, likewise, reward-relevant empowerment as actions that can explicitly modulate future rewards). However, this is just one proposed direction among many possibilities. We suggest one of the most important directions for future work will closely examine how plasticity, empowerment, and their tension relate to goal-directedness more broadly.

**Other Future Work.**    There are many other directions for future work. First, as a consequence of the conditions in Lemma 4.2 and the tension highlighted by Theorem 4.8, there are new opportunities to inform the design of agents. In particular, we speculate that it is prudent to *design agents to balance plasticity and empowerment* tactfully, since over-optimizing one can impact the other. We foresee an opportunity to recover new fundamental results connecting the limits of learning, the design of safe agents, and this balance—if an agent has insufficient plasticity, it cannot adapt, and if it has insufficient empowerment, it cannot steer. Additionally, it may be valuable to better understand how people handle the tension revealed by Theorem 4.8, similar to understanding how people address the explore-exploit dilemma. Second, there is a clear opportunity to make use of our conception of plasticity to formally characterize the stability-plasticity dilemma under minimal assumptions. We envision a clear path to doing so, but defer the full exposition of this narrative to future work. Third, we foresee a direct connection between plasticity, empowerment, and many standard concepts within game theory and multi-agent systems such as the emergence of social dilemmas, cooperation, and the dynamics of multi-agent systems more generally (Jaques et al., 2019). Fourth, a natural next step is to design efficient estimators for the GDI, and by consequence, plasticity and empowerment. Following the work by Jiao et al. (2013), we suspect that a similar algorithmic template can underlie an estimator for GDI. We include a brief experiment to ground our theory in Appendix E, though suggest that deeper empirical work, and an efficient GDI estimator, are valuable directions for further research. Lastly, we speculate that plasticity, empowerment, and their relationship offer a simple, powerful conception of agency: An input-output system is best thought of as an agent if and only if it has non-zero plasticity *and* non-zero empowerment. We again defer a full conceptual analysis of this sort for future work, but refer the reader to the works by Wooldridge and Jennings (1995), Barandiaran et al. (2009), Ball (2023), and Kasirzadeh and Gabriel (2025) for similar discussions.

**Acknowledgments**

The authors are grateful to Pablo Samuel Castro, Claudia Clopath, Iason Gabriel, Joel Z Leibo, Ben Van Roy, and the anonymous reviewers for their thoughtful comments on a draft of the paper. We would further like to thank Barel Skuratovsky and the Agency team for inspirational conversations.

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

# A  Notation

We first provide Table 1 summarizing all relevant notation.

| Notation | Meaning | Definition |
|---|---|---|
| $\mathcal{X}, \mathcal{Y}$ | Set | |
| $X, Y, A, O$ | Discrete random variable | |
| $X_{1:n}$ | Sequence of $n$ random variables | $(X_1, \ldots, X_n)$ |
| $[a:b]_n$ | An interval of discrete time | $[a:b]$ s.t. $1 \le a \le b \le n$ |
| | | |
| $\mathcal{A}$ | Set of actions | |
| $\mathcal{O}$ | Set of observations | |
| $\mathcal{H}_{..\mathcal{A}}$ | Histories ending in action | $(\mathcal{O} \times \mathcal{A})^* \cup (\mathcal{A} \times \mathcal{O})^* \times \mathcal{A}$ |
| $\mathcal{H}_{..\mathcal{O}}$ | Histories ending in observation | $(\mathcal{A} \times \mathcal{O})^* \cup (\mathcal{O} \times \mathcal{A})^* \times \mathcal{O}$ |
| $\mathcal{H}$ | Set of all histories | $\mathcal{H} = \mathcal{H}_{..\mathcal{A}} \cup \mathcal{H}_{..\mathcal{O}}$ |
| | | |
| $e$ | Environment | $e : \mathcal{H}_{..\mathcal{A}} \to \Delta(\mathcal{O})$ |
| $\mathcal{E}_{\mathrm{all}}$ | Set of all environments | $\Delta(\mathcal{O})^{\mathcal{H}_{..\mathcal{A}}}$ |
| $\mathcal{E}$ | Set of Environments | $\mathcal{E} \subseteq \mathcal{E}_{\mathrm{all}}$ |
| | | |
| $\lambda$ | Agent | $\lambda : \mathcal{H}_{..\mathcal{O}} \to \Delta(\mathcal{A})$ |
| $\Lambda_{\mathrm{all}}$ | Set of all agents | $\Delta(\mathcal{A})^{\mathcal{H}_{..\mathcal{O}}}$ |
| $\Lambda$ | Set of agents | $\Lambda \subseteq \Lambda_{\mathrm{all}}$ |
| | | |
| $\mathbb{I}(X_{1:n} \to Y_{1:n})$ | Directed information | $\sum\limits_{i=1:n} \mathbb{I}(X_{1:i}, Y_i \mid Y_{1:i-1})$ |
| $\mathbb{I}(X_{1:n} \hookrightarrow Y_{1:n})$ | Directed information, $X$ delay | $\sum\limits_{i=2:n} \mathbb{I}(X_{1:i-1}; Y_i \mid Y_{1:i-1})$ |
| $\mathbb{I}(X_{a:d} \to Y_{c:d})$ | GDI | $\sum\limits_{i=\max(a,c):d} \mathbb{I}(X_{a:\min(b,i)}; Y_i \mid X_{1:a-1}, Y_{1:i-1})$ |
| | | |
| $\mathfrak{P}_{\substack{a:b \\ c:d}}(\lambda, \mathcal{E})$ | Plasticity | $\max_{e \in \mathcal{E}} \mathbb{I}(O_{a:b} \to A_{c:d})$ |
| $\mathfrak{P}_{\substack{a:b \\ c:d}}(\lambda, e)$ | Plasticity, single environment | $\mathfrak{P}_{\substack{a:b \\ c:d}}(\lambda, \{e\})$ |
| $\mathfrak{P}(\lambda)$ | Plasticity abbreviation | $\mathfrak{P}_{\substack{a:b \\ c:d}}(\lambda, e)$ |
| | | |
| $\mathfrak{E}_{\substack{a:b \\ c:d}}(\Lambda, e)$ | Empowerment | $\max_{\lambda \in \Lambda} \mathbb{I}(A_{a:b} \to O_{c:d})$ |
| $\mathfrak{E}_{\substack{a:b \\ c:d}}(\lambda, e)$ | Empowerment, single agent | $\mathfrak{E}_{\substack{a:b \\ c:d}}(\{\lambda\}, e)$ |
| $\mathfrak{E}(\lambda)$ | Empowerment abbreviation | $\mathfrak{E}_{\substack{a:b \\ c:d}}(\lambda, e)$ |

Table 1: A summary of notation.

# B  Expanded Discussion

Next, we provide additional discussion on salient topics.

## B.1  Optionality, Information, and Causality

Both empowerment and plasticity center around how the agent or environment could have acted or unfolded differently. In the case of empowerment, we ask: Had the agent acted differently, how much could have changed in the environment? For plasticity: If the environment had unfolded differently, how much would that have impacted what the agent did? In this way, both concepts implicitly deal with considerations regarding free will, counterfactuals, causality, and determinism. For example, empowerment is defined by maximizing over the set $\Lambda$, reflecting the different ways an agent *could* respond to stimuli. Constraining $\Lambda$—for instance, by restricting actions, compute, or memory—will in turn reduce the agent's ability to influence future observations and, consequently, its empowerment. In the most general case we can take the max over all agents, $\Lambda_{\mathrm{all}}$, reflecting every possibly way experience could give rise to action over a given fixed interface. However, there are often further restrictions in place, such as the morphology of a robot or a limited available computational budget—such restrictions can be reflected by considering different subsets $\Lambda \subset \Lambda_{\mathrm{all}}$, thereby encoding the different possible ways the agent *could* act. Capdepuy (2011) discusses this distinction in Section 4.1.3, noting that *potential* information flow involves taking the max over available policies, while actual information flow is relative to a single policy. Similarly, plasticity (Definition 4.1) involves maximization over a set of environments $\mathcal{E}$. This choice raises the question: What constitutes environment optionality? How could an environment unfold differently? Interventions, as described by Pearl (2009), offer a formal framework for grounding the notion of environment optionality from a causal perspective. Here, an external experimenter modifies the environment and observes the responses of its sub-systems (such as the agent). This aligns with the continual learning literature, where plasticity is treated as the agent's capacity to adapt to interventions on the data-generating process. For example, in work by Dohare et al. (2021) the input distribution is repeatedly and randomly intervened on, and later work by Dohare et al. (2024) the labels are randomly permuted, whereas in work by Abbas et al. (2023) the agent is switched between different Atari games. These distributional shifts can be formalized as interventions $t$ on a baseline environment $e$, where each intervention creates a new, distributionally shifted environment $e' = t(e)$. In each study there is an assumed set of interventions $\mathcal{T}$ (such as label permutations), which generates the environment set $\mathcal{E} = \{t_i(e) : t_i \in \mathcal{T}\}$. In essence, an agent's plasticity is its capacity to adapt to environmental changes (interventions), defined relative to a set of environments $\mathcal{E}$ generated by these interventions. And, analogous to constraining the policy set for empowerment, plasticity is non-increasing if the set of possible environmental changes is constrained (as in point iii of Theorem 4.3). We suggest that drawing out a full causal picture of both plasticity and empowerment is a compelling direction for further work.

As a consequence of the definition that focuses on observation and action, we find that every deterministic environment forces zero plasticity. That is, if $e$ is deterministic in the sense that it can be written as a map, $e : \mathcal{H}_{..\mathcal{A}} \to \mathcal{O}$, then for every agent and intervals, $\mathfrak{P}_{\substack{a:b \\ c:d}}(\lambda, e) = 0$. This may appear puzzling at first, but on closer inspection, there are several intuitions to elucidate this fact. First, from the agent's point of view, a deterministic environment presents no optionality. This is similar to a causal conception of plasticity relative to the empty set of interventions. Since the environment *could not have been otherwise*, there is no way that the agent can react to the different ways the environment could have unfolded. However, a variant of plasticity that builds around the internal components of the agent (such as the agent's beliefs or memories) will overcome this in a simple way. Consider an agent with limited memory. That is, at most the agent can remember five observations. Then, if the agent is at time-step seven, it cannot possibly remember all prior observations due to its memory constraints. Therefore, from the agent's memory-constrained point of view, the environment may appear stochastic. While these threads play an important conceptual role in plasticity and empowerment, we defer a full exploration of their significance to future work.

## B.2  The Plasticity-Empowerment Tension

Next, we present a short example expanding on the discussion of the tension between plasticity and empowerment highlighted by Theorem 4.8. Consider the example environment depicted in Figure 3.

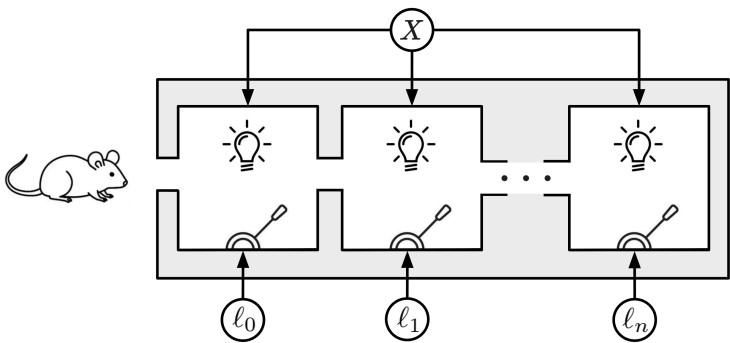

Figure 3: An example environment in which an agent (the mouse) is presented with a gradual change in the balance between their realized plasticity and empowerment. The mouse inhabits a corridor consisting of $n + 1$ rooms, each equipped with a light that can be (stochastically) controlled by the lever in that room.

The environment contains $n + 1$ rooms, each containing a switch and a light. The environment is conditioned on a latent Bernoulli random variable, $X$, corresponding to the probability to turn the light on or off each time step. The mouse only observes whether the light is on or off in the room they currently occupy. The action space of the mouse enables it to move to a neighboring room, or pull the lever. In each room $\ell_i$ a pull of the lever has a probability of $i/n$ to change the state of the light for the next time step, with the remaining probability $n - i/n$ determined by $X$—hence, in room $\ell_0$, the mouse has no control over the light, and in room $\ell_n$, the mouse fully controls whether the light is on or off. Otherwise, the random variable $X$ dictates the state of the light. Suppose $X$ is a fair coin, that is, $X \sim \text{Bernoulli}(\theta = 0.5)$. So, without intervention, the lights in all rooms switch state with probability $0.5$. In this environment, the far left room *maximizes plasticity*, whereas the far right room *maximizes empowerment*, with all rooms in the middle smoothly interpolating between these two extremes. On the left, the mouse gets to receive the most new information from the environment, while on the right, the mouse fully controls the state of the light. Theorem 4.8 suggests that this trade-off is always present to some extent.

To emphasize the tension even further, we can imagine that the latent random variable $X$ is non-stationary. That is, consider the sequence of smoothly evolving Bernoulli random variables, $X_1, X_2, \ldots$. Here, again, if the agent always inhabits the far right room, it fully closes off the possibility of learning about the evolution of $\{X_i\}_{i=1}^{\infty}$. However, if the agent is in the far left room, then the light is fully determined by the evolution of $X_i$, and thus the agent has maximized its potential to learn about the sequence from the environment. Each room in between these two extremes interpolates between these two extremes, roughly corresponding to a walk along the Pareto frontier of the trade-off expressed by the theorem.

## C  Proofs of Presented Results

We next provide proofs of each result from the paper, split into those results presented in Section 3 on GDI (Appendix C.1) and those results presented in Section 4 on plasticity and empowerment (Appendix C.2). Then, in Appendix D, we present additional results on the GDI, including a new data-processing inequality for the GDI.

### C.1  Proofs of Results from Section 3 on GDI

**Proposition 3.2** (GDI Stricly Generalizes DI). *Let $a = c = 1$ and $b = d = n$. Then,*

$$\mathbb{I}(X_{a:b} \to Y_{c:d}) = \mathbb{I}(X_{1:n} \to Y_{1:n}). \tag{20}$$

*Proof of Proposition 3.2.*

The result is a straightforward consequence of the definition of GDI that can be illustrated by substitution. Let $a = c$ and $b = d$ be indices such that $1 = a = c$ and $b = d = n$. Then, rewriting

Equation 10, we find:

$$\mathbb{I}(X_{a:b} \to Y_{c:d}) = \sum_{i=\max(a,c)}^{d} \mathbb{I}(X_{a:\min(b,i)}; Y_i \mid X_{1:a-1}, Y_{1:i-1}), \tag{21}$$

$$= \sum_{i=1}^{n} \mathbb{I}(X_{1:i}; Y_i \mid \cancel{X_{1:i-1}}, Y_{1:i-1}), \tag{22}$$

$$= \underbrace{\sum_{i=1}^{n} \mathbb{I}(X_{1:i}; Y_i \mid Y_{1:i-1})}_{=\mathbb{I}(X_{1:n} \to Y_{1:n})}. \qquad \square \tag{23}$$

**Proposition 3.3** (Temporal Consistency of GDI). *If $a > d$ then $\mathbb{I}(X_{a:b} \to Y_{c:d}) = 0$, and if $a \geq d$ then $\mathbb{I}(X_{a:b} \hookrightarrow Y_{c:d}) = 0$.*

*Proof of Proposition 3.3.*

The property follows directly from the summation bounds of Equation 10, and its variant,

$$\mathbb{I}(X_{a:b} \hookrightarrow Y_{c:d}) \overset{\text{def}}{=} \sum_{i=\max(a+1,c)}^{d} \mathbb{I}(X_{a:\min(b,i-1)}; Y_i \mid X_{1:a-1}, Y_{1:i-1}). \tag{23}$$

For the former, note that if $a > d$, then

$$\mathbb{I}(X_{a:b} \to Y_{c:d}) = \sum_{i=\max(a,c)}^{d} \mathbb{I}(X_{a:\min(b,i)}; Y_i \mid X_{1:a-1}, Y_{1:i-1}) = 0, \tag{24}$$

since $\max(a,c) > d$ when $a > d$.

For the latter, note that if $a \geq d$, then

$$\mathbb{I}(X_{a:b} \hookrightarrow Y_{c:d}) = \sum_{i=\max(a+1,c)}^{d} \mathbb{I}(X_{a:\min(b,i-1)}; Y_i \mid X_{1:a-1}, Y_{1:i-1}) = 0, \tag{25}$$

by similar reasoning that $\max(a+1, c) > d$. $\qquad \square$

**Proposition 3.4** (Summation of Intervals). *Let $a \leq k < b$ and $c \leq \ell < d$.*

$$\mathbb{I}(X_{a:b} \to Y_{c:d}) = \mathbb{I}(X_{a:b} \to Y_{c:\ell}) + \mathbb{I}(X_{a:b} \to Y_{\ell+1:d}) \tag{26}$$

$$= \mathbb{I}(X_{a:k} \to Y_{c:d}) + \mathbb{I}(X_{k+1:b} \to Y_{c:d}). \tag{27}$$

*and similarly,*

$$\mathbb{I}(X_{a:b} \hookrightarrow Y_{c:d}) = \mathbb{I}(X_{a:b} \hookrightarrow Y_{c:f}) + \mathbb{I}(X_{a:b} \hookrightarrow Y_{f+1:d}) \tag{28}$$

$$= \mathbb{I}(X_{a:e} \hookrightarrow Y_{c:d}) + \mathbb{I}(X_{e+1:b} \hookrightarrow Y_{c:d}), \tag{29}$$

*Proof of Proposition 3.4.*

The first equality of Proposition 3.4 consists of breaking the sum of Equation 10 into two sums with intervals $[\max(a,c), \ell]$ and $[\max(a, \ell+1), d]$.

For the second equality observe that if $b > d$ then we can change $b = d$ as $i \not> d$ in the sum of Equation 10. Then, we can apply Equation 10, break up the first sum at $k$, and recombine terms, as follows:

$$\mathbb{I}(X_{a:k} \to Y_{c:d}) + \mathbb{I}(X_{k+1:b} \to Y_{c:d}) \tag{30}$$

$$= \sum_{i=\max(a,c)}^{d} \mathbb{I}(X_{a:\min(k,i)}; Y_i \mid X_{1:a-1}, Y_{1:i-1}) +$$

$$\sum_{i=\max(k+1,c)}^{d} \mathbb{I}(X_{k+1:\min(b,i)}; Y_i \mid X_{1:k}, Y_{1:i-1}), \tag{31}$$

Then, we break the right sum into two intervals,

$$= \sum_{i=\max(a,c)}^{k} \mathbb{I}(X_{a:i}; Y_i \mid X_{1:a-1}, Y_{1:i-1}) + \sum_{i=\max(k+1,c)}^{d} \mathbb{I}(X_{a:k}; Y_i \mid X_{1:a-1}, Y_{1:i-1}) +$$

$$\sum_{i=\max(k+1,c)}^{d} \mathbb{I}(X_{k+1:\min(b,i)}; Y_i \mid X_{1:k}, Y_{1:i-1}). \tag{32}$$

$$= \sum_{i=\max(a,c)}^{k} \mathbb{I}(X_{a:i}; Y_i \mid X_{1:a-1}, Y_{1:i-1}) + \tag{33}$$

$$\sum_{i=\max(k+1,c)}^{d} \left( \mathbb{I}(X_{a:k}; Y_i \mid X_{1:a-1}, Y_{1:i-1}) + \mathbb{I}(X_{k+1:\min(b,i)}; Y_i \mid X_{1:k}, Y_{1:i-1}) \right).$$

Applying the mutual information chain rule to the terms in the second sum and noticing if $i \leq k$ then $i = \min(b, i)$ so we can change the bounds on the first sum to get

$$\mathbb{I}(X_{a:k} \to Y_{c:d}) + \mathbb{I}(X_{k+1:b} \to Y_{c:d}) = \sum_{i=\max(a,c)}^{k} \mathbb{I}(X_{a:\min(b,i)}; Y_i \mid X_{1:a-1}, Y_{1:i-1}) +$$

$$\sum_{i=\max(k+1,c)}^{d} \mathbb{I}(X_{a:\min(b,i)}; Y_i \mid X_{1:a-1}, Y_{1:i-1}). \tag{34}$$

Now we combine the sums to get our result,

$$\mathbb{I}(X_{a:k} \to Y_{c:d}) + \mathbb{I}(X_{k+1:b} \to Y_{c:d})$$

$$= \sum_{i=\max(a,c)}^{d} \mathbb{I}(X_{a:\min(b,i)}; Y_i \mid X_{1:a-1}, Y_{1:i-1}) = \mathbb{I}(X_{a:b} \to Y_{c:d}). \quad \square \tag{35}$$

To prove the conservation law of GDI, we first introduce a short lemma that helps decompose the GDI into a difference of entropies.

Next, we present a further decomposition of any GDI term that will be a key step in our proof of the conservation law of GDI.

**Lemma C.1.** *Let* $[a : b]_n, [c : d]_n$ *be valid intervals, and let* $k = \max(a, c), \ell = \min(b, d)$. *Then,*

$$\mathbb{I}(X_{a:b} \to Y_{c:d}) \tag{36}$$
$$= \mathbb{I}(X_{a:k-1}; Y_{k:\ell} \mid X_{1:a-1}, Y_{1:k-1}) + \mathbb{I}(X_{k:\ell} \to Y_{k:\ell}) + \mathbb{I}(X_{a:\ell}; Y_{\ell+1:d} \mid X_{1:a-1}, Y_{1:\ell}),$$

*and*

$$\mathbb{I}(Y_{c:d} \hookrightarrow X_{a:b}) \tag{37}$$
$$= \mathbb{I}(Y_{c:k-1}; X_{k:\ell} \mid Y_{1:c-1}, X_{1:k-1}) + \mathbb{I}(Y_{k:\ell} \hookrightarrow X_{k:\ell}) + \mathbb{I}(Y_{c:\ell}; X_{\ell+1:b} \mid Y_{1:c-1}, X_{1:\ell}).$$

*Proof of Lemma C.1.*

Applying Proposition 3.4 multiple times,

$$\mathbb{I}(X_{a:b} \to Y_{c:d}) \tag{38}$$

$$
\begin{aligned}
= \quad & \mathbb{I}(X_{a:k-1} \to Y_{c:k-1}) &+& \quad \mathbb{I}(X_{k:\ell} \to Y_{c:k-1}) &+& \quad \mathbb{I}(X_{\ell+1:b} \to Y_{c:k-1}) &+& \\
& \mathbb{I}(X_{a:k-1} \to Y_{k:\ell}) &+& \quad \mathbb{I}(X_{k:\ell} \to Y_{k:\ell}) &+& \quad \mathbb{I}(X_{\ell+1:b} \to Y_{k:\ell}) &+& \\
& \mathbb{I}(X_{a:k-1} \to Y_{\ell+1:d}) &+& \quad \mathbb{I}(X_{k:\ell} \to Y_{\ell+1:d}) &+& \quad \mathbb{I}(X_{\ell+1:b} \to Y_{\ell+1:d}). &&
\end{aligned}
$$

Notice that either $k = a$ or $k = b$, and similarly $\ell = c$ or $\ell = d$, so two terms are guaranteed to have empty ranges,

$$\mathbb{I}(X_{a:b} \to Y_{c:d}) \tag{39}$$

$$
\begin{aligned}
= \quad & \cancel{\mathbb{I}(X_{a:k-1} \to Y_{c:k-1})} &+& \quad \mathbb{I}(X_{k:\ell} \to Y_{c:k-1}) &+& \quad \mathbb{I}(X_{\ell+1:b} \to Y_{c:k-1}) &+& \\
& \mathbb{I}(X_{a:k-1} \to Y_{k:\ell}) &+& \quad \mathbb{I}(X_{k:\ell} \to Y_{k:\ell}) &+& \quad \mathbb{I}(X_{\ell+1:b} \to Y_{k:\ell}) &+& \\
& \mathbb{I}(X_{a:k-1} \to Y_{\ell+1:d}) &+& \quad \mathbb{I}(X_{k:\ell} \to Y_{\ell+1:d}) &+& \quad \cancel{\mathbb{I}(X_{\ell+1:b} \to Y_{\ell+1:d})}. &&
\end{aligned}
$$

We can now apply Proposition 3.3 to remove additional terms.

$$\mathbb{I}(X_{a:b} \to Y_{c:d}) \tag{40}$$

$$
\begin{aligned}
= \quad & \cancel{\mathbb{I}(X_{a:k-1} \to Y_{c:k-1})} &+& \quad \cancel{\mathbb{I}(X_{k:\ell} \to Y_{c:k-1})} &+& \quad \cancel{\mathbb{I}(X_{\ell+1:b} \to Y_{c:k-1})} &+& \\
& \mathbb{I}(X_{a:k-1} \to Y_{k:\ell}) &+& \quad \mathbb{I}(X_{k:\ell} \to Y_{k:\ell}) &+& \quad \cancel{\mathbb{I}(X_{\ell+1:b} \to Y_{k:\ell})} &+& \\
& \mathbb{I}(X_{a:k-1} \to Y_{\ell+1:d}) &+& \quad \mathbb{I}(X_{k:\ell} \to Y_{\ell+1:d}) &+& \quad \cancel{\mathbb{I}(X_{\ell+1:b} \to Y_{\ell+1:d})}. &&
\end{aligned}
$$

Recombining the last two terms with Proposition 3.4 gives us three remaining terms,

$$\mathbb{I}(X_{a:b} \to Y_{c:d}) = \mathbb{I}(X_{a:k-1} \to Y_{k:\ell}) + \mathbb{I}(X_{k:\ell} \to Y_{k:\ell}) + \mathbb{I}(X_{a:\ell} \to Y_{\ell+1:d}) \tag{41}$$

Applying Equation 10 to the first and last term,

$$\mathbb{I}(X_{a:b} \to Y_{c:d}) \tag{42}$$

$$= \sum_{i=k}^{\ell} \mathbb{I}(X_{a:k-1}; Y_i \mid X_{1:a-1}, Y_{1:i-1}) + \mathbb{I}(X_{k:\ell} \to Y_{k:\ell}) + \sum_{i=\ell+1}^{d} \mathbb{I}(X_{a:\ell}; Y_i \mid X_{1:a-1}, Y_{1:i-1}),$$

and applying the mutual information chain rule to these two summations gives,

$$\mathbb{I}(X_{a:b} \to Y_{c:d}) \tag{43}$$

$$= \mathbb{I}(X_{a:k-1}; Y_{k:\ell} \mid X_{1:a-1}, Y_{1:k-1}) + \mathbb{I}(X_{k:\ell} \to Y_{k:\ell}) + \mathbb{I}(X_{a:\ell}; Y_{\ell+1:d} \mid X_{1:a-1}, Y_{1:\ell}).$$

This completes the first case. The second case follows by similar reasoning,

$$
\begin{aligned}
& \mathbb{I}(Y_{c:d} \hookrightarrow X_{a:b}) \\
= \quad & \cancel{\mathbb{I}(Y_{c:k-1} \hookrightarrow X_{a:k-1})} &+& \quad \cancel{\mathbb{I}(Y_{k:\ell} \hookrightarrow X_{a:k-1})} &+& \quad \cancel{\mathbb{I}(Y_{\ell+1:d} \hookrightarrow X_{a:k-1})} &+& \\
& \cancel{\mathbb{I}(Y_{c:k-1} \hookrightarrow X_{k:\ell})} &+& \quad \cancel{\mathbb{I}(Y_{k:\ell} \hookrightarrow X_{k:\ell})} &+& \quad \cancel{\mathbb{I}(Y_{\ell+1:d} \hookrightarrow X_{k:\ell})} &+& \\
& \mathbb{I}(Y_{c:k-1} \hookrightarrow X_{\ell+1:b}) &+& \quad \mathbb{I}(Y_{k:\ell} \hookrightarrow X_{\ell+1:b}) &+& \quad \cancel{\mathbb{I}(Y_{\ell+1:d} \hookrightarrow X_{\ell+1:b})} && \\
= \quad & \mathbb{I}(Y_{c:k-1} \hookrightarrow X_{k:\ell}) + \mathbb{I}(Y_{k:\ell} \hookrightarrow X_{k:\ell}) + \mathbb{I}(Y_{c:\ell} \hookrightarrow X_{\ell+1:d}) \\
= \quad & \mathbb{I}(Y_{c:k-1}; X_{k:\ell} \mid Y_{1:c-1}, X_{1:k-1}) + \mathbb{I}(Y_{k:\ell} \hookrightarrow X_{k:\ell}) + \mathbb{I}(Y_{c:\ell}; X_{\ell+1:b} \mid Y_{1:c-1}, X_{1:\ell})
\end{aligned}
$$

This completes both cases, and we conclude. $\qquad\square$

**Theorem 3.5** (Conservation Law of GDI).

$$\mathbb{I}(X_{a:b}; Y_{c:d} \mid X_{1:a-1}, Y_{1:c-1}) = \mathbb{I}(X_{a:b} \to Y_{c:d}) + \mathbb{I}(Y_{c:d} \hookrightarrow X_{a:b}). \tag{44}$$

*Proof of Theorem 3.5.*

The proof proceeds by decomposing the two GDI terms $\mathbb{I}(X_{a:b} \to Y_{c:d})$ and $\mathbb{I}(Y_{c:d} \hookrightarrow X_{a:b})$ by application of Lemma C.1, then inducting on $\ell - k$ as in the original proof by Massey and Massey (2005).

First, we decompose both $\mathbb{I}(X_{a:b} \to Y_{c:d})$ and $\mathbb{I}(Y_{c:d} \hookrightarrow X_{a:b})$ using Lemma C.1.

$$\mathbb{I}(X_{a:b} \to Y_{c:d}) \tag{45}$$
$$= \mathbb{I}(X_{a:k-1}; Y_{k:\ell} \mid |X_{1:a-1}, Y_{1:k-1}) + \mathbb{I}(X_{k:\ell} \to Y_{k:\ell}) + \mathbb{I}(X_{a:\ell}; Y_{\ell+1:d} \mid X_{1:a-1}, Y_{1:\ell}),$$

and

$$\mathbb{I}(Y_{c:d} \hookrightarrow X_{a:b}) \tag{46}$$
$$= \mathbb{I}(Y_{c:s-1}; X_{k:\ell} \mid Y_{1:c-1}, X_{1:k-1}) + \mathbb{I}(Y_{k:\ell} \hookrightarrow X_{k:\ell}) + \mathbb{I}(Y_{c:\ell}; X_{\ell+1:b} \mid Y_{1:c-1}, X_{1:\ell}).$$

Let us now consider $\mathbb{I}(X_{k:\ell} \to Y_{k:\ell}) + \mathbb{I}(Y_{k:\ell} \hookrightarrow X_{k:\ell})$, which is almost the term constrained by the law of conservation of directed information (when $k = 1$). We use induction on $\ell - k$ as employed by Massey and Massey (2005) to show the sum is $\mathbb{I}(X_{k:\ell}; Y_{k:\ell} \mid X_{1:k-1}, Y_{1:k-1})$. First, if $\ell = k$ then $\mathbb{I}(X_{k:\ell} \to Y_{k:\ell}) = \mathbb{I}(X_{k:\ell}; Y_{k:\ell} \mid X_{1:k-1}, Y_{1:k-1})$ and $\mathbb{I}(Y_{k:\ell} \hookrightarrow X_{k:\ell}) = 0$, thus establishing the base case. In the general case,

$$\mathbb{I}(X_{k:\ell} \to Y_{k:\ell}) + \mathbb{I}(Y_{k:\ell} \hookrightarrow X_{k:\ell}) = \begin{aligned}&\mathbb{I}(X_{k:\ell-1} \to Y_{k:\ell-1}) + \mathbb{I}(X_{k:\ell}; Y_e \mid X_{1:k-1}, Y_{1:\ell-1}) + \\ &\mathbb{I}(Y_{k:\ell-1} \hookrightarrow X_{k:\ell-1}) + \mathbb{I}(X_e; Y_{k:\ell-1} \mid X_{1:\ell-1}, Y_{1:k-1}).\end{aligned}$$

By rearranging and applying induction,

$$\begin{aligned}= \quad &\mathbb{I}(X_{k:\ell-1}; Y_{k:\ell-1} \mid X_{1:k-1}, Y_{1:k-1}) + \\ &\mathbb{I}(X_\ell; Y_{k:\ell-1} \mid X_{1:\ell-1}, Y_{1:k-1}) + \mathbb{I}(X_{k:\ell}; Y_e \mid X_{1:k-1}, Y_{1:\ell-1}),\end{aligned} \tag{47}$$

Then, applying the mutual information chain rule on the first two terms,

$$= \quad \mathbb{I}(X_{k:\ell}; Y_{k:\ell-1} \mid X_{1:k-1}, Y_{1:k-1}) + \mathbb{I}(X_{k:\ell}; Y_\ell \mid X_{1:k-1}, Y_{1:\ell-1}), \tag{48}$$

and once more on the remaining terms,

$$= \quad \mathbb{I}(X_{k:\ell}; Y_{k:\ell} \mid X_{1:k-1}, Y_{1:k-1}), \tag{49}$$

Thus by induction,

$$\mathbb{I}(X_{k:\ell} \to Y_{k:\ell}) + \mathbb{I}(Y_{k:\ell} \hookrightarrow X_{k:\ell}) = \mathbb{I}(X_{k:\ell}; Y_{k:\ell} \mid X_{1:k-1}, Y_{1:k-1}). \tag{50}$$

We now combine Equation 45, Equation 46, and Equation 50,

$$\mathbb{I}(X_{a:b} \to Y_{c:d}) + \mathbb{I}(Y_{c:d} \hookrightarrow X_{a:b}) = \mathbb{I}(X_{a:k-1}; Y_{k:\ell} \mid X_{1:a-1}, Y_{1:k-1}) + \tag{51}$$
$$\mathbb{I}(X_{k:\ell}; Y_{c:s-1} \mid X_{1:k-1}, Y_{1:c-1}) + \mathbb{I}(X_{k:\ell}; Y_{k:\ell} \mid X_{1:k-1}, Y_{1:k-1}) + \tag{52}$$
$$\mathbb{I}(X_{a:\ell}; Y_{\ell+1:d} \mid X_{1:a-1}, Y_{1:\ell}) + \mathbb{I}(X_{\ell+1:b}; Y_{c:\ell} \mid X_{1:\ell}, Y_{1:c-1}). \tag{53}$$

Notice that at most one of the first two terms is non-zero (if $a > c$ the top term is zero; if $a < c$ the bottom term is zero), and similarly for the final two terms (if $b > d$ the top term is zero; if $b < d$ the bottom term is zero). We now apply the mutual information chain rule to the non-zero first term with the middle term. In either case we get,

$$\mathbb{I}(X_{a:b} \to Y_{c:d}) + \mathbb{I}(Y_{c:d} \hookrightarrow X_{a:b}) \tag{54}$$
$$= \mathbb{I}(X_{a:\ell}; Y_{c:\ell} \mid X_{1:a-1}, Y_{1:b-1}) + \mathbb{I}(X_{a:\ell}; Y_{\ell+1:d} \mid X_{1:a-1}, Y_{1:\ell}) + \tag{55}$$
$$\mathbb{I}(X_{\ell+1:b}; Y_{c:\ell} \mid X_{1:\ell}, Y_{1:c-1}). \tag{56}$$

We can now apply the mutual chain rule with the non-zero last term. In either case, we get,

$$\mathbb{I}(X_{a:b} \to Y_{c:d}) + \mathbb{I}(Y_{c:d} \hookrightarrow X_{a:b}) = \mathbb{I}(X_{a:b}; Y_{c:d} \mid X_{1:a-1}, Y_{1:b-1}), \tag{57}$$

which confirms the identity, and we conclude the proof. $\square$

## C.2 Proofs of Results from Section 4 on Plasticity and Empowerment

**Lemma 4.2** (Necessary and Sufficient Conditions for Positive Plasticity). *Agents have non-zero plasticity relative to $e$ if and only if there is a time-step where their actions are influenced by observation.*

*More formally, for any pair $(\lambda, e)$ and indices $[a:b]_n$, $[c:d]_n$, there exists an $i \in [\max(a,c):d]$ such that*

$$\mathbb{H}(A_i \mid O_{1:a-1}, A_{1:i-1}) > \mathbb{H}(A_i \mid O_{a:\min(b,i)}, O_{1:a-1}, A_{1:i-1}) \tag{58}$$

*if and only if $\mathfrak{P}_{\substack{a:b\\c:d}}(\lambda, e) > 0$.*

**Proof of Lemma 4.2.**

We first establish necessity: If $\mathfrak{P}_{\substack{a:b\\c:d}}(\lambda, e) > 0$, then there exists an $\exists_{i' \in [\max(a,c):d]}$ such that

$$\mathbb{H}(A_{i'} \mid O_{1:a-1}, A_{1:i'-1}) > \mathbb{H}(A_{i'} \mid O_{a:\min(b,i')}, O_{1:a-1}, A_{1:i'-1}). \tag{59}$$

Expanding the definition of $\mathfrak{P}_{\substack{a:b\\c:d}}(\lambda, e)$,

$$\mathfrak{P}_{\substack{a:b\\c:d}}(\lambda, e) = \mathbb{I}(O_{a:b} \to A_{c:d}) \tag{60}$$

$$= \sum_{i=\max(a,c)}^{d} \mathbb{I}(O_{a:\min(b,i)}; A_i \mid O_{1:a-1}, A_{1:i-1}). \tag{61}$$

Recall again that the conditional mutual information is non-negative. Hence, the sum

$$\sum_{i=\max(a,c)}^{d} \mathbb{I}(O_{a:\min(b,i)}; A_i \mid O_{1:a-1}, A_{1:i-1}), \tag{62}$$

is only greater than zero if at least one of its terms is non-zero. Since $\mathfrak{P}_{\substack{a:b\\c:d}}(\lambda, e) > 0$, we conclude that there must be an $i' \in [\max(a,c):d]$ such that $\mathbb{I}(O_{a:\min(b,i')}; A_{i'} \mid O_{1:a-1}, A_{1:i'-1}) > 0$. We next apply the identity that conditional mutual information can be expressed as a difference of conditional entropy terms,

$$\underbrace{\mathbb{H}(A_{i'} \mid O_{1:a-1}, A_{1:i'-1}) - \mathbb{H}(A_{i'} \mid O_{a:\min(b,i')}, O_{1:a-1}, A_{1:i'-1})}_{\mathbb{I}(O_{a:\min(b,i')}; A_i \mid O_{1:a-1}, A_{1:i'-1})} > 0, \tag{63}$$

and therefore,

$$\mathbb{H}(A_{i'} \mid O_{1:a-1}, A_{1:i'-1}) > \mathbb{H}(A_{i'} \mid O_{a:\min(b,i')}, O_{1:a-1}, A_{1:i'-1}). \tag{64}$$

This completes the argument for necessity. $\checkmark$

We next prove sufficiency: If there exists an $i' \in [\max(a,c):d]$ such that

$$\mathbb{H}(A_{i'} \mid O_{1:a-1}, A_{1:i'-1}) > \mathbb{H}(A_{i'} \mid O_{a:\min(b,i')}, O_{1:a-1}, A_{1:i'-1}), \tag{65}$$

then $\mathfrak{P}_{\substack{a:b\\c:d}}(\lambda, e) > 0$.

The same reasoning from the necessity argument may be applied in reverse. First, we can rewrite the above inequality and replace the difference with the conditional mutual information,

$$\mathbb{I}(O_{a:\min(b,i')}; A_{i'} \mid O_{1:a-1}, A_{1:i'-1}) > 0, \tag{66}$$

which in turn implies that

$$\sum_{i=\max(a,c)}^{d} \mathbb{I}(O_{a:\min(b,i)}; A_i \mid O_{1:a-1}, A_{1:i-1}) > 0, \tag{67}$$

and by the definition of plasticity, we conclude $\mathfrak{P}_{\substack{a:b\\c:d}}(\lambda, e) > 0$. $\square$

**Theorem 4.3.** *The following properties hold of the function $\mathfrak{P}$, for all intervals $[a:b]_n$, $[c:d]_n$:*

*(i) For all $(\lambda, \mathcal{E})$, $\mathfrak{P}_{\substack{a:b\\c:d}}(\lambda, \mathcal{E}) \geq 0$.*

*(ii) For all $a < d$, there exists a pair $(\lambda, \mathcal{E})$ where $\lambda$ is deterministic and $\mathfrak{P}_{\substack{a:b\\c:d}}(\lambda, \mathcal{E}) > 0$.*

*(iii) For all $\mathcal{E}_{\text{small}} \subseteq \mathcal{E}_{\text{big}}$, $\mathfrak{P}_{\substack{a:b\\c:d}}(\lambda, \mathcal{E}_{\text{small}}) \leq \mathfrak{P}_{\substack{a:b\\c:d}}(\lambda, \mathcal{E}_{\text{big}})$.*

*(iv) (Informal) The following agents all have zero plasticity relative to every environment set: Closed-loop agents, agents that always output the same action distribution, agents whose actions only depend on history length, and agents whose actions only depend on past actions.*

**Proof of Theorem 4.3.**

We prove each property independently.

*(i) For all $(\lambda, \mathcal{E})$ and any intervals $[a:b]_n$, $[c:d]_n$, $\mathfrak{P}_{\substack{a:b \\ c:d}}(\lambda, \mathcal{E}) \geq 0$.*

This follows as a direct consequence of the first inequality of Proposition D.2, together with the definition of plasticity.      ✓

*(ii) For all $a < d$, there exists a pair $(\lambda, \mathcal{E})$ where $\lambda$ is deterministic and $\mathfrak{P}_{\substack{a:b \\ c:d}}(\lambda, \mathcal{E}) > 0$.*

It suffices to construct a single deterministic agent with non-zero plasticity for a fixed but arbitrary pair of intervals where $a < d$. Consider the singleton environment set containing $e$ that outputs a uniform distribution on $\mathcal{O}$ at each time-step.

To be precise, we define a deterministic agent as any non-stochastic mapping from histories to actions, $\lambda : \mathcal{H}_{..\mathcal{O}} \to \mathcal{A}$, so the set of all such deterministic agents is the set of all such mappings,

$$\Lambda_{\text{det}} = \mathcal{A}^{\mathcal{H}_{..\mathcal{O}}}. \tag{68}$$

Then, let $\lambda \in \Lambda_{\text{det}}$ be a deterministic function that outputs $a_1$ if the last observation was $o_1$, and outputs $a_2$ otherwise. That is,

$$\lambda(ho) = \begin{cases} \delta\{a_1\} & o = o_1, \\ \delta\{a_2\} & \text{otherwise,} \end{cases} \tag{69}$$

for all $h \in \mathcal{H}_{..\mathcal{A}}, o \in \mathcal{O}$, where $\delta\{a_i\}$ is the Dirac delta distribution on $a_i$.

Then, for any choice of intervals where $a < d$, we see that for any $i \in [\max(a,c):d]$,

$$\underbrace{\mathbb{H}(A_i \mid O_{1:a-1}, A_{1:i-1})}_{>0} > \underbrace{\mathbb{H}(A_i \mid O_{a:\min(b,i)}, O_{1:a-1}, A_{1:i-1})}_{=0}. \tag{70}$$

Again by application of Lemma 4.2, this implies that $\mathfrak{P}_{\substack{a:b \\ c:d}}(\lambda, e) > 0$, and we conclude.      ✓

*(iii) For all $\mathcal{E}_{\text{small}} \subseteq \mathcal{E}_{\text{big}}$ and any intervals $[a:b]_n, [c:d]_n$, $\mathfrak{P}_{\substack{a:b \\ c:d}}(\lambda, \mathcal{E}_{\text{small}}) \leq \mathfrak{P}_{\substack{a:b \\ c:d}}(\lambda, \mathcal{E}_{\text{big}})$.*

The result follows directly from the use of the max operator in the definition of plasticity. That is, since

$$\max_{x \in \mathcal{X}_{\text{small}}} f(x) \leq \max_{x \in \mathcal{X}_{\text{big}}} f(x), \tag{71}$$

for any sets $\mathcal{X}_{\text{small}} \subseteq \mathcal{X}_{\text{big}}$ and function $f$, then

$$\underbrace{\max_{e \in \mathcal{E}_{\text{small}}} \mathbb{I}(O_{a:b} \to A_{c:d})}_{=\mathfrak{P}_{\substack{a:b \\ c:d}}(\lambda, \mathcal{E}_{\text{small}})} \leq \underbrace{\max_{e \in \mathcal{E}_{\text{big}}} \mathbb{I}(O_{a:b} \to A_{c:d})}_{=\mathfrak{P}_{\substack{a:b \\ c:d}}(\lambda, \mathcal{E}_{\text{big}})}, \tag{72}$$

as desired.      ✓

*(iv) (Informal) The following agents all have zero plasticity relative to every environment set: Agents that always output the same action distribution, agents whose actions only depend on history length, and agents whose actions only depend on past actions.*

The argument follows directly for all three agent classes by application of Lemma 4.2: For each agent type, conditioning on observations tells us nothing about the actions the agent emits.

To be precise, we define a constant agent as an element of the set

$$\Lambda_{\text{constant}} = \{\lambda \in \Lambda : \lambda(h) = \lambda(h'), \forall_{h,h' \in \mathcal{H}_{..\mathcal{O}}}\}. \tag{73}$$

That is, a constant agent outputs the same probability distribution over actions in every history.

For any such constant agent, we find that

$$\mathbb{H}(A_i \mid O_{1:a-1}, A_{1:i-1}) = \mathbb{H}(A_i \mid O_{a:\min(b,i)}, O_{1:a-1}, A_{1:i-1})), \tag{74}$$

since each $A_i \sim \lambda_{\mathrm{c}}(H)$ is conditionally independent of $O_{a:\min(b,i)}$ given $A_{1:a-1}, O_{1:i-1}$, by definition of $\lambda_{\mathrm{c}}$ as a constant agent. By Lemma 4.2, this concludes the argument.

The same reasoning applies to agents whose actions only depend on the length of history, closed-loop agents, and agents whose actions only depend on past actions.

This concludes proof of all four properties. □

**Proposition 4.5.** $\mathfrak{E}_{\substack{1:n \\ 1:n}}(\Lambda, e) = \mathfrak{E}_{\mathrm{C}}^n(\Lambda, e)$ *and* $\mathfrak{E}_{\substack{1:n \\ n:n}}(\Lambda_{a^n}, e) = \mathfrak{E}_{\mathrm{K}}^n(\Lambda_{a^n}, e)$.

*Proof of Proposition 4.5.*

We show that Capdepuy's and Klyubin et al.'s definitions of empowerment are special cases in separate arguments.

(Capdepuy)

First, we can see that the GDI-based definition captures Capdepuy's definition when $1 = a = c$ and $b = d = n$, since

$$\mathfrak{E}_{\substack{a:b \\ c:d}}(\Lambda, e) = \max_{\lambda \in \Lambda} \mathbb{I}(A_{1:n} \to O_{1:n}) = \mathfrak{E}_{\mathrm{C}}^n(\Lambda, e). \qquad \checkmark \tag{75}$$

(Klyubin et al.)

Second, we show that when $a = 1$, $c = n$, and $b = d = n$ and the set of agents in question are the open loop agents, $\Lambda_{a^n}$, then

$$\mathfrak{E}_{\substack{a:b \\ c:d}}(\Lambda_{a^n}, e) = \mathfrak{E}_{\substack{1:n \\ n:n}}(\Lambda_{a^n}, e) = \mathfrak{E}_{\mathrm{K}}^n(\Lambda_{a^n}, e), \tag{76}$$

We start by expanding the GDI-based definition,

$$\mathfrak{E}_{\substack{1:n \\ n:n}}(\Lambda_{a^n}, e) = \max_{\lambda \in \Lambda_{a^n}} \mathbb{I}(A_{a:b} \to O_{c:d}), \tag{77}$$

$$= \max_{\lambda \in \Lambda} \mathbb{I}(A_{1:n} \to O_{n:n}), \tag{78}$$

$$= \max_{p(a^n) \in \Lambda_{a^n}} \mathbb{I}(A_{1:n} \to O_n). \tag{79}$$

Now, since the agents are open loop, we know that $\mathbb{I}(O_n \hookrightarrow A_{1:n}) = 0$. By Proposition 2.3,

$$\underbrace{\mathbb{I}(O_n \hookrightarrow A_{1:n})}_{=0} + \mathbb{I}(A_{1:n} \to O_n) = \mathbb{I}(A_{1:n}; O_n), \tag{80}$$

and hence,

$$\mathbb{I}(A_{1:n} \to O_n) = \mathbb{I}(A_{1:n}; O_n). \tag{81}$$

Therefore,

$$\mathfrak{E}_{\substack{a:b \\ c:d}}(\Lambda_{a^n}, e) = \max_{p(a^n) \in \Lambda_{a^n}} \mathbb{I}(A_{1:n} \to O_n) \tag{82}$$

$$= \max_{p(a^n) \in \Lambda_{a^n}} \mathbb{I}(A_{1:n}; O_n) \tag{83}$$

$$= \mathfrak{E}_{\mathrm{K}}^n(\Lambda_{a^n}, e). \checkmark \tag{84}$$

This conclude both equalities, and we conclude the proof. □

**Proposition 4.6** (Plasticity is the Mirror of Empowerment)**.** *Given any pair $(\lambda, e)$ that share an interface, and valid intervals $[a : b]_n, [c : d]_n$, the agent's empowerment is equal to the plasticity of*

*the environment, and the agent's plasticity is equal to the empowerment of the environment:*

$$\mathfrak{E}_{\substack{a:b\\c:d}}(\lambda) = \mathfrak{P}_{\substack{a:b\\c:d}}(e), \qquad \mathfrak{P}_{\substack{a:b\\c:d}}(\lambda) = \mathfrak{E}_{\substack{a:b\\c:d}}(e). \tag{85}$$

***Proof of Proposition 4.6.***

The proof proceeds as a direct consequence of the definitions of empowerment and plasticity, and the symmetry highlighted by Remark 2.8.

First, we expand each definition for the agent,

$$\mathfrak{P}_{\substack{a:b\\c:d}}(\lambda) = \mathfrak{P}_{\substack{a:b\\c:d}}(\lambda, e) = \mathbb{I}(O_{a:b} \to A_{c:d}), \tag{86}$$

$$\mathfrak{E}_{\substack{a:b\\c:d}}(\lambda) = \mathfrak{E}_{\substack{a:b\\c:d}}(\lambda, e) = \mathbb{I}(A_{a:b} \to O_{c:d}). \tag{87}$$

Now, to arrive at analogous definitions for the environment, we exploit the symmetry highlighted by Remark 2.8 to recover notions of plasticity and empowerment when the environment is treated as an agent,

$$\mathfrak{P}_{\substack{a:b\\c:d}}(e) = \mathfrak{P}_{\substack{a:b\\c:d}}(e, \lambda) = \mathbb{I}(A_{a:b} \to O_{c:d}), \tag{88}$$

$$\mathfrak{E}_{\substack{a:b\\c:d}}(e) = \mathfrak{E}_{\substack{a:b\\c:d}}(e, \lambda) = \mathbb{I}(O_{a:b} \to A_{c:d}). \tag{89}$$

Now, notice that $\mathfrak{E}(\lambda) = \mathfrak{P}(e)$ by Equation 87 and Equation 88, and $\mathfrak{P}(\lambda) = \mathfrak{E}(e)$ by Equation 86 and Equation 89. $\qquad\square$

**Theorem 4.8.** *For all intervals $[a:b]_n, [c:d]_n$ and any pair $(\lambda, e)$, let $m = \min\{(b - a + 1)\log|\mathcal{O}|, (d - c + 1)\log|\mathcal{A}|\}$. Then,*

$$\mathfrak{E}_{\substack{a:b\\c:d}}(\lambda, e) + \mathfrak{P}_{\substack{c:d\\a:b}}(\lambda, e) \leq m, \tag{90}$$

*is a tight upper bound on empowerment and plasticity. Furthermore, under mild assumptions on the interface and interval, there exists a pair $(\lambda^\diamond, e^\diamond)$ such that $\mathfrak{P}_{\substack{c:d\\a:b}}(\lambda^\diamond, e^\diamond) = m$, and there exists a pair $(\lambda', e')$ such that $\mathfrak{E}_{\substack{a:b\\c:d}}(\lambda', e') = m$, thereby forcing the other quantity to be zero.*

***Proof of Theorem 4.8.***

Let $(\lambda, e)$ be a fixed but arbitrary agent-environment pair, $[a:b]_n, [c:d]_n$ be valid intervals, and let $m = \min\{(b - a + 1)\log|\mathcal{O}|, (d - c + 1)\log|\mathcal{A}|\}$.

Then, by Theorem 3.5, we arrive at the identity,

$$\mathfrak{P}_{\substack{c:d\\a:b}}(\lambda, e) + \mathfrak{E}_{\substack{a:b\\c:d}}(\lambda, e) =_1 \mathbb{I}(O_{a:b}; A_{c:d} \mid O_{1:a-1}, A_{1:c-1}), \tag{91}$$

$$\leq_2 \min\left\{\mathbb{H}(O_{a:b}), \mathbb{H}(A_{c:d})\right\}, \tag{92}$$

$$\leq_3 \min\{(b - a + 1)\log|\mathcal{O}|, (d - c + 1)\log|\mathcal{A}|\}, \tag{93}$$

$$= m, \tag{94}$$

where $=_1$ follows from the conservation law of GDI (Theorem 3.5), $\leq_2$ follows by the standard upper bounds on conditional mutual information, and $\leq_3$ follows from the standard upper bound on joint entropy.

To prove this upper bound is tight, we show there exists two pairs, $(\lambda^\diamond, e^\diamond)$ and $(\lambda', e')$ such that

$$m - \mathfrak{P}_{\substack{a:b\\c:d}}(\lambda^\diamond, e^\diamond) = 0, \qquad m - \mathfrak{E}_{\substack{a:b\\c:d}}(\lambda', e') = 0, \tag{95}$$

We present the argument for plasticity, and recall that by symmetry (Remark 2.8), the same will hold of empowerment.

To do so, we require that $|\mathcal{A}| \leq |\mathcal{O}|$ and $(b - a) \geq (d - c)$. While the desired inequality will likely hold under softer conditions, the argument is cleanest under these conditions. For simplicity, we also assume $a \leq c$.

By Proposition D.3, we express $\mathfrak{P}_{\substack{a:b\\c:d}}(\lambda^\diamond, e^\diamond)$ as follows

$$\mathfrak{P}_{\substack{a:b\\c:d}}(\lambda^\diamond, e^\diamond) = \mathbb{I}(O_{a:b} \to A_{c:d}) \tag{96}$$

$$= \mathbb{H}(A_{\max(a,c):d} \mid O_{1:a-1}) - \mathbb{H}(A_{c:d} \mid\mid O_{a:b}). \tag{97}$$

Since we assumed $a \leq c$, we rewrite

$$\mathfrak{P}_{\substack{a:b \\ c:d}}(\lambda^\diamond, e^\diamond) = \mathbb{H}(A_{c:d} \mid O_{1:a-1}) - \mathbb{H}(A_{c:d} \parallel O_{a:b}). \tag{98}$$

Let $k = |\mathcal{A}|$, and let $\mathcal{O}_k$ refer to the set containing the first $k$ observations, $\mathcal{O}_k = \{o_1, o_2, \ldots, o_k\}$. Let $e^\diamond$ be the environment that always emits $o_1$ along the initial (possibly empty) interval $[1 : a-1]_n$, and emits a uniform random distribution over $\mathcal{O}_k$ from time $i = a$ onward. Now, let $\lambda^\diamond$ be the agent that always plays $a_1$ until the start of the observation interval $[a : b]$, then on seeing any observation $o_j$, it mirrors the same action back, $a_j$. More formally,

$$e(h_i) = \begin{cases} \delta\{o_1\} & i \in [1 : a-1], \\ \text{Unif}(\mathcal{O}_k) & i \geq a, \end{cases} \qquad \lambda(h_i o_j) = \begin{cases} \delta\{a_1\} & i < a \lor j > k, \\ \delta\{a_j\} & \text{otherwise}, \end{cases} \tag{99}$$

for all $h_i \in \mathcal{H}_{..\mathcal{A}}$, $o_j \in \mathcal{O}$, where $h_i o_j \in \mathcal{H}_{..\mathcal{O}}$ expresses the concatenation of $h_i$ and $o_j$. Additionally, we let $\delta\{x_i\}$ express the Dirac delta distribution on $x_i$.

In this situation, observe that

$$\mathbb{H}(A_{c:d} \mid O_{1:a-1}) = (d - c) \cdot \log |\mathcal{A}|, \tag{100}$$

since the agent plays the uniform random distribution over $\mathcal{A}$ in this interval by mirroring the environment's uniform random distribution over $\mathcal{O}_k$, and we assumed $(b - a) \geq (d - c)$. Furthermore,

$$\mathbb{H}(A_{c:d} \parallel O_{a:b}) = 0 \tag{101}$$

since learning about the observation at each time-step in the interval $[a : b]$ fully determines the action in the interval $[c : c + (b - a)]$. Therefore, substituting into Equation 98

$$\mathfrak{P}_{\substack{a:b \\ c:d}}(\lambda^\diamond, e^\diamond) = \underbrace{\mathbb{H}(A_{c:d} \mid O_{1:a-1})}_{\geq m} - \underbrace{\mathbb{H}(A_{c:d} \parallel O_{a:b})}_{=0}. \tag{102}$$

Hence, we conclude that

$$\mathfrak{P}_{\substack{a:b \\ c:d}}(\lambda^\diamond, e^\diamond) \geq m, \tag{103}$$

and by the non-negativity of empowerment and plasticity,

$$\mathfrak{E}_{\substack{a:b \\ c:d}}(\lambda^\diamond, e^\diamond) \leq m - \mathfrak{P}_{\substack{a:b \\ c:d}}(\lambda^\diamond, e^\diamond) = 0. \tag{104}$$

This completes the argument for the case of empowerment, and we call attention to the fact that by symmetry (Remark 2.8), an identical argument holds for plasticity. $\qquad\square$

# D   Other Results

**Theorem D.1** (Data-processing inequality for GDI). *Consider any tuple $(X_{1:n}, Y_{1:n}, Z_{1:n})$ where $Z_i$ is conditionally independent of $X_{1:i}$ given $Y_i$, for each $i = 1 \ldots n$. Then,*

$$\mathbb{I}(X_{a:b} \to Y_{c:d}) \geq \mathbb{I}(X_{a:b} \to Z_{c:d}), \tag{105}$$

*for any valid intervals $[a : b]_n, [c : d]_n$.*

***Proof of Theorem D.1.***

The argument follows an identical structure to the standard argument for the data-processing inequality for regular mutual information, together with two inductive steps. The first inductive step applies to the interval $[a : b]_n$ and the second applies to the interval $[c : d]_n$. We assume across all arguments that each $Z_i$ is conditionally independent of $X_{1:i}$ given $Y_i$ for each $i = 1 \ldots n$.

Base Case: $a = b, c = d$

First, we consider the base case when $a = b$ and $c = d$, and assume $a \leq d$ (if $a > d$, we know $\mathbb{I}(X_{a:b} \to Y_{c:d}) = 0$ by Proposition 3.3). Hence, we write $\mathbb{I}(X_{a:a} \to Y_{c:c}) = \mathbb{I}(X_a \to Y_c)$.

Expanding the definition for both relations $X$ to $Y$ and $X$ to $Z$, we fine,

$$\mathbb{I}(X_a \to Y_c) = \sum_{i=\max(a,c)}^{d} \mathbb{I}(X_{a:\min(b,i)}; Y_i \mid X_{1:a-1}Y_{1:i-1}) = \mathbb{I}(X_a; Y_{\max(a,c)}), \qquad (106)$$

$$\mathbb{I}(X_a \to Z_c) = \sum_{i=\max(a,c)}^{d} \mathbb{I}(X_{a:\min(b,i)}; Z_i \mid X_{1:a-1}Z_{1:i-1}) = \mathbb{I}(X_a; Z_{\max(a,c)}). \qquad (107)$$

Then, we apply the standard proof technique for the data-processing inequality: We consider the mutual information $\mathbb{I}(X_a; Y_{\max(a,c)}, Z_{\max(a,c)})$, and use the identity that

$$\mathbb{I}(X_a; Y_{\max(a,c)}, Z_{\max(a,c)}) = \mathbb{I}(X_a; Y_{\max(a,c)}) + \mathbb{I}(X_a; Y_{\max(a,c)} \mid Z_{\max(a,c)}), \qquad (108)$$

$$= \mathbb{I}(X_a; Z_{\max(a,c)}) + \underbrace{\mathbb{I}(X_a; Z_{\max(a,c)} \mid Y_{\max(a,c)})}_{=0}. \qquad (109)$$

Where the final term is equal to zero by the assumption that each $Z_i$ is conditionally independent of $X_{1:i}$ given $Y_i$ for each $i = 1 \dots n$. Therefore, rearranging, we find that

$$\mathbb{I}(X_a; Y_{\max(a,c)}) + \mathbb{I}(X_a; Y_{\max(a,c)} \mid Z_{\max(a,c)}) = \mathbb{I}(X_a; Z_{\max(a,c)}), \qquad (110)$$

and by the non-negativity of mutual information, we conclude

$$\underbrace{\mathbb{I}(X_a; Y_{\max(a,c)})}_{=\mathbb{I}(X_{a:a} \to Y_{c:c})} \geq \underbrace{\mathbb{I}(X_a; Z_{\max(a,c)})}_{=\mathbb{I}(X_{a:a} \to Z_{c:c})}, \qquad (111)$$

as desired. This concludes the base case. ✓

We next turn to the two inductive cases.

Inductive Case 1: $\mathbb{I}(X_{a:b} \to Y_{c:c}) \geq \mathbb{I}(X_{a:b} \to Z_{c:c})$.

We assume as our inductive hypothesis that

$$\mathbb{I}(X_{a:b-1} \to Y_{c:c}) \geq \mathbb{I}(X_{a:b-1} \to Z_{c:c}), \qquad (112)$$

and show that this implies

$$\mathbb{I}(X_{a:b} \to Y_{c:c}) \geq \mathbb{I}(X_{a:b} \to Z_{c:c}). \qquad (113)$$

We apply Proposition 3.4 once on the terms $\mathbb{I}(X_{a:b} \to Y_{c:c})$ and $\mathbb{I}(X_{a:b} \to Z_{c:c})$ to decompose them into

$$\mathbb{I}(X_{a:b} \to Y_{c:c}) = \mathbb{I}(X_{a:b-1} \to Y_{c:c}) + \mathbb{I}(X_{b:b} \to Y_{c:c}) \qquad (114)$$

$$\mathbb{I}(X_{a:b} \to Z_{c:c}) = \mathbb{I}(X_{a:b-1} \to Z_{c:c}) + \mathbb{I}(X_{b:b} \to Z_{c:c}). \qquad (115)$$

But then, by the inductive hypothesis and the base case, we know that

$$\mathbb{I}(X_{a:b-1} \to Y_{c:c}) \geq \mathbb{I}(X_{a:b-1} \to Z_{c:c}), \qquad \mathbb{I}(X_{b:b} \to Y_{c:c}) \geq \mathbb{I}(X_{b:b} \to Z_{c:c}). \qquad (116)$$

Thus,

$$\underbrace{\mathbb{I}(X_{a:b} \to Y_{c:c})}_{=\mathbb{I}(X_{a:b-1} \to Y_{c:c}) + \mathbb{I}(X_{b:b} \to Y_{c:c})} \geq \underbrace{\mathbb{I}(X_{a:b} \to Z_{c:c})}_{=\mathbb{I}(X_{a:b-1} \to Z_{c:c}) + \mathbb{I}(X_{b:b} \to Z_{c:c})}, \qquad (117)$$

as desired. ✓

Inductive Case 2: $\mathbb{I}(X_{a:b} \to Y_{c:d}) \geq \mathbb{I}(X_{a:b} \to Z_{c:d})$.

The argument for this inductive case is identical to the former, only we apply the inductive step to the interval $[c:d]_n$. ✓

This concludes the proof. □

Next, as with most terms related to mutual information, the GDI is non-negative and upper bounded.

**Proposition D.2** (GDI Upper Bound).

$$0 \leq \mathbb{I}(X_{a:b} \to Y_{c:d}) \leq \mathbb{I}(X_{a:b}; Y_{c:d} \mid X_{1:a-1}, Y_{1:c-1}). \tag{118}$$

*Proof of Proposition D.2.*

The two inequalities follow as a consequence of the definition of GDI, and its conservation law (Theorem 3.5).

First, note that $\mathbb{I}(X_{a:b} \to Y_{c:d})$ is non-negative by definition:

$$\mathbb{I}(X_{a:b} \to Y_{c:d}) = \sum_{i=\max(a,c)}^{d} \underbrace{\mathbb{I}(X_{a:\min(b,i)}; Y_i \mid X_{1:a-1}, Y_{1:i-1})}_{\geq 0}. \tag{119}$$

since it is the sum of conditional mutual information terms that are each non-negative by definition. Second, by Theorem 3.5, we know that

$$\mathbb{I}(X_{a:b} \to Y_{c:d}) + \mathbb{I}(Y_{c:d} \hookrightarrow X_{a:b}) = \mathbb{I}(X_{a:b}; Y_{c:d} \mid X_{1:a-1}, Y_{1:c-1}), \tag{120}$$

and since all three terms are non-negative, it follows that

$$\mathbb{I}(X_{a:b} \to Y_{c:d}) \leq \mathbb{I}(X_{a:b}; Y_{c:d} \mid X_{1:a-1}, Y_{1:c-1}). \qquad \square$$

Lastly, we can decompose the GDI into conditional entropy terms.

**Proposition D.3** (GDI Kramer Decomposition).

$$\mathbb{I}(X_{a:b} \to Y_{c:d}) = \mathbb{H}(Y_{\max(a,c):d} \mid X_{1:a-1}) - \mathbb{H}(Y_{c:d} \mid\mid X_{a:b}), \tag{121}$$

*where*

$$\mathbb{H}(Y_{c:d} \mid\mid X_{a:b}) = \sum_{i=\max(a,c)}^{d} \mathbb{H}(Y_i \mid Y_{1:i-1}, X_{a:\min(b,i)}), \tag{122}$$

*is the generalization of Kramer's causal entropy to arbitrary indices.*

*Proof of Proposition D.3.*

The result follows by expanding the definition of GDI and applying the conditional mutual information identity in terms of a difference of conditional entropy terms, as follows.

$$\mathbb{I}(X_{a:b} \to Y_{c:d}) = \sum_{i=\max(a,c)}^{d} \mathbb{I}(X_{a:\min(b,i)}; Y_i \mid X_{1:a-1}, Y_{1:i-1}), \tag{123}$$

$$= \sum_{i=\max(a,c)}^{d} \mathbb{H}(Y_i \mid X_{1:a-1}, Y_{1:i-1}) - \mathbb{H}(Y_i \mid X_{a:\min(b,i)}, X_{1:a-1}, Y_{1:i-1}),$$

$$= \sum_{i=\max(a,c)}^{d} \mathbb{H}(Y_i \mid X_{1:a-1}, Y_{1:i-1}) - \mathbb{H}(Y_i \mid Y_{1:i-1}, X_{1:\min(b,i)}), \tag{124}$$

$$= \sum_{i=\max(a,c)}^{d} \mathbb{H}(Y_i \mid X_{1:a-1}, Y_{1:i-1}) - \underbrace{\sum_{i=\max(a,c)}^{d} \mathbb{H}(Y_i \mid Y_{1:i-1}, X_{1:\min(b,i)})}_{=\mathbb{H}(Y_{c:d} \mid\mid X_{a:b})},$$

$$= \underbrace{\left( \sum_{i=\max(a,c)}^{d} \mathbb{H}(Y_i \mid X_{1:a-1}, Y_{1:i-1}) \right)}_{=\mathbb{H}(Y_{\max(a,c):d} \mid X_{1:a-1})} - \mathbb{H}(Y_{c:d} \mid\mid X_{a:b}), \tag{125}$$

$$= \mathbb{H}(Y_{\max(a,c):d} \mid X_{1:a-1}) - \mathbb{H}(Y_{c:d} \mid\mid X_{a:b}). \qquad \square$$

As a direct corollary of Proposition D.3 together with Proposition 3.2, we recover Kramer's identity as a special case, as follows.

**Corollary D.4.** *When* $1 = a = c$ *and* $b = d = n$,

$$\mathbb{H}(Y_{\max(a,c):d} \mid X_{1:a-1}) - \mathbb{H}(Y_{c:d} \parallel X_{a:b}) = \mathbb{H}(Y_{1:n}) - \mathbb{H}(Y_{1:n} \parallel X_{1:n}), \tag{126}$$

*and,*

$$\mathbb{H}(Y_{c:d} \parallel X_{a:b}) = \mathbb{H}(Y_{1:n} \parallel X_{1:n}). \tag{127}$$

# E  Experiments

Lastly, we conduct two simple experiments to ground and corroborate our theory. In each case, we make use of a basic Monte Carlo estimator for the GDI, and study the plasticity and empowerment of tabular $Q$-learning (Watkins and Dayan, 1992) interacting with a two-armed Bernoulli bandit. The experiments were run on a single CPU.

In the first experiment, we use an $\epsilon$-greedy policy for exploration and study the impact of varying $\epsilon$ between $[0, 1]$ on plasticity. We estimate the plasticity of $Q$-learning for each of these values of $\epsilon$ in the time intervals $[1 : 3]$ and $[2 : 5]$, so $\mathbb{I}(O_{1:3} \to A_{2:5})$. Results are shown in Figure 4(a). We see that, when $\epsilon = 0$ the agent is relying entirely on the greedy policy to drive its actions, which is determined by the past observations. Thus, with a low $\epsilon$, we see a higher level of plasticity. Conversely, as we increase $\epsilon$, the degree to which the greedy policy drives action choice decreases. At the maximal value of $\epsilon = 1$, we find that plasticity is zero since the agent's actions are no longer influenced by observation.

In the second experiment, we study the impact of optimism and pessimism on plasticity and empowerment, again in the two-armed Bernoulli bandit. To vary the degree of optimism or pessimism present in the agent, we vary the initial $Q$ value used for each action from $-1$ to $1$ and examine the impact on plasticity and empowerment along the same intervals $[1 : 3]$ and $[2 : 5]$. Here, we no longer use $\epsilon$-greedy exploration, and rely solely on the initial $Q$ value to drive exploration. Results are shown in Figure 4(b). First, we note that the upper-bound (depicted in grey) from Theorem 4.8 is corroborated empirically. Second, we see that empowerment tends to be higher than plasticity in this setting—the agent influences the environment more than the environment influences the agent. Third, we find that empowerment increases with optimism. This likely comes about due to the degree of exploration present in the agent: When the initial $Q$ values are pessimistic (strictly below zero), the agent will be more likely to stick with the first arm that delivers positive reward. In this way, the pessimistic agent is less incentivized to try out different actions, which leads to a lower overall empowerment.

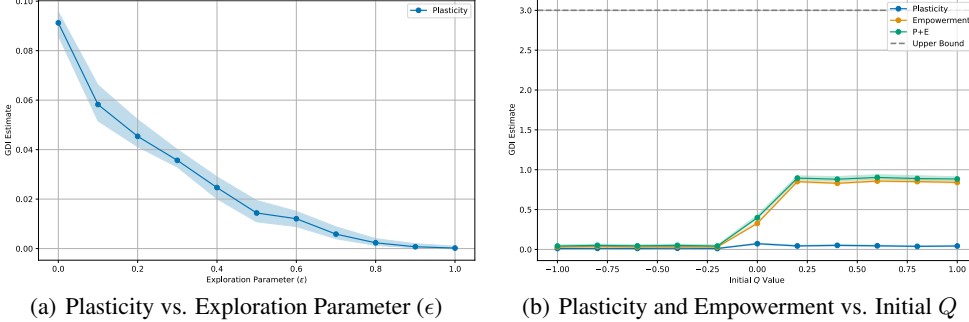

(a) Plasticity vs. Exploration Parameter ($\epsilon$)   (b) Plasticity and Empowerment vs. Initial $Q$

Figure 4: Results from two simple experiments that estimate plasticity, empowerment, and their sum with tabular $Q$-learning (Watkins and Dayan, 1992) in a two-armed bandit. On the left, we show the mean estimate of an $\epsilon$-greedy variant of $Q$-learning as a function of $\epsilon$, varied from zero to one, with bootstrapped confidence intervals. On the right, we show the estimate of plasticity (blue), empowerment (orange), and their sum (green), along with the upper bound (grey-dashed line) as a function of the initial $Q$ value used by $Q$-learning, ranging from negative one to one. When the initial $Q = -1$, the agent is pessimistic, and when the initial $Q = 1$, the agent is optimistic.

