# OpenReview forum: "Plasticity as the Mirror of Empowerment"
_NeurIPS.cc/2025/Conference — NeurIPS 2025 spotlight_

### Official Review · Reviewer_o8k4 · 2025-06-30

**Clarity:** 4
**Significance:** 3
**Originality:** 3
**Rating:** 5
**Confidence:** 3

**Summary:**

This paper notes that in the abstract, and an agent and an environment are symmetric communication partners; we just call the agent's messages "actions", and the environment's messages "observations". Following, and slightly extending, the literature on "empowerment", the paper observes that the empowerment of the environment is an interesting quantity that corresponds to an agent's "plasticity"--the extent to which the environment can guide its behavior. Ultimately, the paper proves an upper bound on the sum of an agent's plasticity and its empowerment, indicating that the two (desirable) quantities are in tension with each other.

**Questions:**

Could you please explain to me why Theorem 4.8 holds?

**Ethical Concerns:**

["NO or VERY MINOR ethics concerns only"]

**Final Justification:**

Thank you to the authors and other reviewers for the discussion. My questions have been answered and I am keeping my score the same.

**Limitations:**

It's hard for me to identify any limitations with this work.

**Quality:**

3

**Strengths And Weaknesses:**

I'll remember this paper, and what I've learned from it. The paper was clear and easy to read, and it is original. It introduces a new conceptual tool, and I think it may be significant.

A lot of the paper consists of developing notation and definitions--I suppose that's necessary for this kind of work. I somehow feel like I didn't gain that much more from reading the paper than I did from reading the abstract, with one important exception, which is Theorem 4.8. So I would have liked to see a proof sketch, and maybe a full proof in the body of the paper.

---

> ### Author Rebuttal · Authors · 2025-07-30
>
> We would like to thank the reviewer for taking their time and energy to read our paper. We really appreciate their thoughtful comments on our work. We have included a brief response to their primary request below, but are happy to engage and discuss any further questions or comments that might come up.
>
> > Could you please explain to me why Theorem 4.8 holds?
>
> We are happy to, and we will plan to make adjustments to the exposition of the result in order to further clarify.
>
> The short intuition is this: if an agent fully controls its environment, it cannot also learn from it. Consider a setting involving a two player band that consists of a guitar player (G) and violin player (V). If G is maximally empowered, it means that the notes they play fully determine what V plays. So, imagine V repeats back exactly the note they hear from G, then V is fully plastic. In this way, G cannot then learn anything meaningful from what V plays, since G will just be hearing exactly what they already played. To get the most out of the collaboration, we likely want G and V to both be a little plastic and a little empowered: they should listen to their collaborator (plasticity), but also try to contribute something meaningful (empowerment).
>
>
> > So I would have liked to see a proof sketch, and maybe a full proof in the body of the paper.
>
> We completely agree with this point, and will plan to include the proof sketch and expanded intuition for the main two results (the GDI conservation and the Plasticity-Empowerment tension) in the camera ready. If we have space, we will be sure to include a proof sketch of this result in the main paper (along with a sketch of the proof of the conservation law of GDI).
>
> We thank the reviewer for their thoughtful review, and are happy to discuss anything further.

---

> > ### Comment · Reviewer_o8k4 · 2025-08-01
> >
> > Thank you!
> >
> > I have read the other reviews, and I am still comfortable with my assessment that this is paper should be accepted.
> >
> > In particular, I agree with the authors' response to this comment from Reviewer 2PEP:
> >
> > > A primary concern about this work is that it seems to be redefining an existing and widely used term “plasticity” to be rather different from the (intuitive but not mathematically precise) typical definition of “an agent’s capacity to remain adaptive”. In this work, the term “plasticity” is instead better described as capturing “the influence of an environment on the agent” and it’s not clear why we should be using the same word ‘plasticity’ for this measure.
> >
> > I'd add that "if an agent has the capacity to remain adaptive", what could actually cause it to adapt? The only route for such an adaptation to be realized is as a result of input from its environment, because the observations produced by the environment are the only inputs to the agent's policy. In that case, the ability for the environment to influence the agent is the same thing as the agent's capacity to adapt.

---

> > > ### Author Response · Authors · 2025-08-03
> > > **Re: Official Comment by Reviewer o8k4**
> > >
> > > Thank you for your comments: We appreciate your thoughtful response, and your close read of the other reviews. We fully agree with your comment regarding the relationship between adaptivity and plasticity. We are very happy to discuss these elements (or any other points) further

---

### Official Review · Reviewer_vgU6 · 2025-07-01

**Clarity:** 4
**Significance:** 4
**Originality:** 4
**Rating:** 6
**Confidence:** 4

**Summary:**

The authors introduce a general definition of "plasticity" for agents. To do so, they create a generalized version of directed information that does not require comparing sequences of equal length from the same time step. They show that agent plasticity is the "mirror" of environment empowerment (and vice versa). Moreover, they show that plasticity and empowerment are in tension: it is not possible to simultaneously maximize both the agent's plasticity and empowerment. This raises questions into what the optimal trade-off is between the two. At a minimum, the authors posit that an agent should have non-zero plasticity and non-zero empowerment.

**Questions:**

I'm curious, what do the authors think is the most important empirical work to do in this space to build upon their proposed conceptualization of plasticity and the plasticity-empowerment trade-off?

**Ethical Concerns:**

["NO or VERY MINOR ethics concerns only"]

**Final Justification:**

I am keeping my already high score (strong accept), for the reasons described above in "strengths and weaknesses"

**Limitations:**

Yes

**Quality:**

4

**Strengths And Weaknesses:**

Overall, I thought this was an _excellent_ paper, and I strongly recommend its acceptance.
- It proposes a conceptually crisp and general notion of "plasticity" that relies on only minimal assumptions on the interaction between agent and environment.
- The definition fits well into existing literature on empowerment as they show that plasticity can be seen as the mirror of empowerment, which is conceptually very appealing
- They provide a generalization of directed information that is needed to study either plasticity or empowerment over time at different intervals
- Their main theoretical results around (a) that agent plasticity = environment empowerment (and vice versa) and (b) the tension between agent plasticity + agent empowerment raise many interesting questions regarding the design of agents. How should plasticity and empowerment be balanced?
- I think this framework will have wide appeal and is broadly relevant for many areas ranging from RL theory to modern LLM agents to recommender systems
- The paper is very well-written and clear

---

> ### Author Rebuttal · Authors · 2025-07-30
>
> __Overview.__ We would like to thank the reviewer for their thoughtful review of our paper, and for their warm comments on the work. We truly appreciate it! We respond to the reviewer’s question below, but if they have added comments or questions during the discussion period we are happy to talk further.
>
> > what do the authors think is the most important empirical work to do in this space to build upon their proposed conceptualization of plasticity and the plasticity-empowerment trade-off?
>
> This is an excellent question, and one we spent some time considering since submission. We have since implemented a small-scale experiment to corroborate the basic findings of the work, but believe there are several important directions to explore to build on the work.
>
> First, we believe it is important to better understand which algorithmic knobs can directly control an agent’s plasticity and empowerment. For example, changing an agent’s learning rate parameter intuitively controls the agent’s plasticity—is there an equivalent, typical parameter that modulates an agent’s empowerment? If not, can we design one? Do these knobs also have knock-on effects to other parts of the agent (does changing learning rate impact empowerment?)
>
> Second, it is important to better understand how goals fit into our picture. As the reviewer indicated in their review, we are curious about what an optimal balance of plasticity and empowerment looks like. To study this question, we might introduce goals in the form of a reward signal as is typical in reinforcement learning, then explore when agents should lean into maximising plasticity, empowerment, or achieving a balanced trade-off. We believe that a systematic empirical study of standard learning algorithms and environments would be a natural place to seek this understanding, particularly in a continual learning setting.
>
> Third, there are opportunities to connect to the multi-agent and game theory literature. In a two-player game, one player’s plasticity is the other player’s empowerment (as in the poker example). In an N-player game, one player’s empowerment contributes to the other players’ plasticities. As such, we believe it would be fruitful to again conduct an experimental study involving traditional solution concepts from the point of view of plasticity and empowerment—for certain kinds of games or dilemmas, what must hold of an equilibrium? What do standard methods achieve in terms of plasticity and empowerment? Can we use the tools of plasticity and empowerment to make predictions about such equilibria?
>
> Lastly, we foresee valuable pathways to connecting to the safety literature. As Turner et al. (2021) explored in _Optimal Policies Tend to Seek Power_, we speculate that one way to modulate the safety of different systems is through the use of tools like plasticity and empowerment. For example, in a collaborative setting, it would be desirable for agents to act in a way to respect the agency of its collaborators—that is, to ensure that it does not systematically damage the plasticity or empowerment of collaborators. We can again imagine an empirical study exploring how algorithms in a cooperative setting might influence the plasticity or empowerment of their collaborators, and try to reveal potential mechanisms for improving these methods.
>
> These are a few of the directions we believe would be valuable to follow up on. Thank you again for the question, and for the time spent reviewing our paper.
>
> References:
> * Turner, A., Smith, L., Shah, R., Critch, A., & Tadepalli, P. (2021). Optimal Policies Tend To Seek Power. Advances in Neural Information Processing Systems, 34.

---

> > ### Comment · Reviewer_vgU6 · 2025-08-05
> >
> > Thank you to the authors for your response and good luck with these directions. Re the safety connections, I agree that testing these concepts out in cooperative settings would be very interesting. Cooperative IRL [1] was also designed with the intention to maintain plasticity of the AI agent / empowerment of the human, with the idea that by making the AI agent's objective the same as the human's (unknown) reward function, it always has incentive to learn from the human. However, even in a cooperative game, one would expect that the AI agent's plasticity decreases over time as it becomes more certain about the human's reward function. I agree that the author's formalism could help shed light, more granularly, on what conditions (e.g. on the environment, algorithm, temporality) lead to which trade-offs between plasticity / empowerment.
> >
> > I will maintain my (already high) score. I have read the authors' rebuttals to the other reviewers and agree with the authors' points.
> >
> > [1] https://arxiv.org/abs/1606.03137

---

> > > ### Author Response · Authors · 2025-08-05
> > > **Re: Official Comment by Reviewer vgU6**
> > >
> > > We thank the reviewer for their kind response, and their additional comments regarding the potential application of these ideas to cooperative settings. We completely agree that exploring connections to Cooperative IRL is a very fruitful direction for future work! We are happy to discuss any other questions or comments that should arise.

---

### Official Review · Reviewer_x8QU · 2025-07-02

**Clarity:** 2
**Significance:** 2
**Originality:** 2
**Rating:** 3
**Confidence:** 3

**Summary:**

This paper proposes a new agent-centric measure (which they call as plasticity) and it quantifies how much an agent is influenced by its observations, complementing the concept of empowerment, which measures the agent’s influence on future observations. Plasticity is formalized using generalized directed information... it is defined by extending Massey’s directed information framework. The authors show two key results:
(a) Plasticity and empowerment are mirror quantities between the agent and environment, and
(b) There exists a trade-off.. both cannot be maximized simultaneously. This reveals a fundamental tension between being shaped by the world and shaping it.
The work offers a fresh theoretical perspective for understanding and designing intelligent agents.

**Questions:**

Please answer below questions:

(a) This paper is entirely theoretical... it does not perform any empirical evaluation to demonstrate the practical implications of their proposed GDI framework. Any reason for not doing this? Is there any lack of data or other issues/

(b) The use of advanced information-theoretic constructs like GDI may hinder accessibility and adoption among practitioners. Are these constructs lead to any engineering constraints?

(c) The agent-environment interaction model is highly abstract and it may not capture the dynamic and uncertain nature of real-world environments where assumptions like full observability or stationarity may not hold. Any comments on this?

(d) While the empowerment–plasticity tradeoff is theoretically intriguing, the paper lacks actionable insights on how to balance or optimize these properties when designing real agents. Do you have any quick suggestions to address this?

**Ethical Concerns:**

["NO or VERY MINOR ethics concerns only"]

**Limitations:**

(a) This paper is entirely theoretical... it does not perform any empirical evaluation to demonstrate the practical implications of their proposed GDI framework. Any reason for not doing this? Is there any lack of data or other issues/

(b) The use of advanced information-theoretic constructs like GDI may hinder accessibility and adoption among practitioners. Are these constructs lead to any engineering constraints?

(c) The agent-environment interaction model is highly abstract and it may not capture the dynamic and uncertain nature of real-world environments where assumptions like full observability or stationarity may not hold. Any comments on this?

(d) While the empowerment–plasticity tradeoff is theoretically intriguing, the paper lacks actionable insights on how to balance or optimize these properties when designing real agents. Do you have any quick suggestions to address this?

**Quality:**

2

**Strengths And Weaknesses:**

Below are my  comments and concerns on this paper:

(a) This paper is entirely theoretical... it does not perform any empirical evaluation to demonstrate the practical implications of their proposed GDI framework. Any reason for not doing this? Is there any lack of data or other issues/

(b) The use of advanced information-theoretic constructs like GDI may hinder accessibility and adoption among practitioners. Are these constructs lead to any engineering constraints?

(c) The agent-environment interaction model is highly abstract and it may not capture the dynamic and uncertain nature of real-world environments where assumptions like full observability or stationarity may not hold. Any comments on this?

(d) While the empowerment–plasticity tradeoff is theoretically intriguing, the paper lacks actionable insights on how to balance or optimize these properties when designing real agents. Do you have any quick suggestions to address this?

---

> ### Author Rebuttal · Authors · 2025-07-30
>
> __Overview.__ We would like to thank the reviewer for their time in reviewing our paper, and for their questions. We respond to each of their four main questions (a-d) in detail below, and will plan to make adjustments to the paper to clarify these points where appropriate.
>
> &nbsp;
> ### Question (a): Theory
>
> > (a) This paper is entirely theoretical... it does not perform any empirical evaluation to demonstrate the practical implications of their proposed GDI framework. Any reason for not doing this? Is there any lack of data or other issues/
>
> Thanks for the question. We do maintain that the theory we have introduced stands on its own as a valuable contribution. However, since submission, we have implemented estimators for plasticity and empowerment, and conducted small-scale experiments that validate and expand on some of the findings of the work. Our first experiment involves a simple class of agents and environments and corroborates the upper bound of Theorem 4.8. Our second experiment involves modulating various characteristics of an agent (we use UCB and Q-learning), such as the degree of optimism or exploration parameters of the agent. Concretely, we adjust things like the initial value function, learning rate, and the exploration parameters of the agents and examine the impact of these changes on plasticity and empowerment. Depending on space, we are happy to include a section about these experiments in the final version of the paper. However, we suggest that a full-scale experimental study that significantly expands the work is beyond our present scope, but agree that it is a useful direction for future research.
>
> &nbsp;
> ### Question (b): GDI
>
> > (b) The use of advanced information-theoretic constructs like GDI may hinder accessibility and adoption among practitioners. Are these constructs lead to any engineering constraints?
>
> If we understand the question correctly, we do not see any obvious ways in which the use of things like the GDI will lead to any engineering constraints. As indicated above, we have implemented simple estimators for these quantities—since all of the measurements are behavioral, we believe that a larger scale experiment is in scope for future work. Information theoretic quantities are commonly used in practice, and there is no reason the GDI (as a generalization of directed information), should raise any new challenges.
>
> &nbsp;
> ### Question (c): Environment Limitations
>
> > (c) The agent-environment interaction model is highly abstract and it may not capture the dynamic and uncertain nature of real-world environments where assumptions like full observability or stationarity may not hold. Any comments on this?
>
> To clarify, the environment model we use is general enough to capture both partial observability and non-stationarity, and includes things like POMDPs and non-stationarity MDPs as special cases. See the work by Dong et al (2022) or Majeed and Hutter (2018) for further information on this generality. Similarly, the agent model we make use of is also highly general, and can capture arbitrary agents.
>
> In this way, we take our theory to be a general one—we make only the mild assumptions that the action and observation spaces are finite: the main results hold for effectively the most general classes of agent and environment types, and believe this is a significant strength of the work. We are happy to emphasize this generality more in the paper.
>
> References:
> * Dong, S., Van Roy, B., & Zhou, Z. (2022). Simple agent, complex environment: Efficient reinforcement learning with agent states. Journal of Machine Learning Research, 23(255), 1-54.
> * Majeed, S. J., & Hutter, M. (2018). On Q-learning Convergence for Non-Markov Decision Processes. IJCAI.
>
> &nbsp;
> ### Question (d): Actionable Insights
>
> > (d) While the empowerment–plasticity tradeoff is theoretically intriguing, the paper lacks actionable insights on how to balance or optimize these properties when designing real agents. Do you have any quick suggestions to address this?
>
> There are several directions we are eager to explore, though we first emphasize that this paper is laying down the foundations that we strongly believe can later influence and materialize in practical guidance of the kind suggested. Our answer here is similar in spirit to Reviewer vGU6’s question:
>
> __1. Algorithm Design.__ First, we believe it is important to better understand which algorithmic knobs can directly control an agent’s plasticity and empowerment. For example, changing an agent’s learning rate parameter intuitively controls the agent’s plasticity—is there an equivalent, typical parameter that modulates an agent’s empowerment? If not, can we design one? Do these knobs also have knock-on effects to other parts of the agent (does changing the learning rate impact empowerment?) Once such knobs are identified, we can leverage them to modulate an agent’s degree of plasticity and empowerment, as indicated by the reviewer. In cases where we want to ensure agents remain adaptive and pliable (as in cases of coordination or continual learning), we can control these knobs to ensure plasticity is maintained. Similarly, if we want to limit the empowerment of an agent (as in safety-critical cases), we can again make use of such knobs.
>
> __2. Goals, Plasticity, Empowerment.__ Second, it is important to better understand how goals fit into our picture. If we were to introduce goals in the form of reward as is typical in reinforcement learning, then it is useful to clarify when agents should lean into maximizing plasticity, empowerment, or achieving a balanced trade-off between the two in order to satisfy their goals. We have so far proven some initial results that clarify the relationship between optimal learning, plasticity, and empowerment, and believe these kinds of results can shed light on when and how we should approach the balance of plasticity and empowerment. We believe that a systematic empirical study of standard learning algorithms and environments would further be a natural place to expand on this understanding, particularly in a continual learning setting.
>
>
>
>
> __3. Balancing Plasticity and Empowerment.__ Third, we note that many have previously argued that agents should maximize their empowerment. For example, in one of the early pioneering works on empowerment, Klyubin et al. (2005) explicitly suggest “All else being equal be empowered”, as their title indicates. Empowerment is then often treated as an optimization objective in its own right (see, for example, work by Leibfried et al., 2019, among many others). Our theory suggests otherwise: if an agent is always maximizing empowerment, it will lower its ability to receive information from and be influenced by the environment (or other agents), too. For example, if an artificial agent is playing a cooperative game with a person, then to cooperate effectively, both the agent and person should be slightly plastic enough to allow themselves to be influenced by their teammate—if they were only empowered, they would look to control their teammate, which is undesirable.
>
> In this way, our theory suggests that maximizing empowerment is not always ideal, but instead, agent design requires balancing plasticity and empowerment tactfully. We believe this represents a significant change in perspective, and one that can inspire future research on the design of cooperative learning agents that act in general environments. While the specifics of how to best achieve this are hinted at above in (1.) and (2.), we would like to emphasize that a principled change in perspective is also valuable.
>
> References:
> * Klyubin, A. S., Polani, D., & Nehaniv, C. L. (2005, September). All else being equal be empowered.
> * Leibfried, F., Pascual-Diaz, S., & Grau-Moya, J. (2019). A unified bellman optimality principle combining reward maximization and empowerment. Advances in Neural Information Processing Systems, 32.
>
>
> __4. Game theory and MARL.__ Fourth, there are opportunities to connect to the multi-agent and game theory literature, and to use multi-agent concepts to motivate the balance of plasticity and empowerment. In a two-player game, one player’s plasticity is the other player’s empowerment (as in the poker example). In an N-player game, one player’s empowerment contributes to the other players’ plasticities. As such, we believe it would be fruitful to again conduct an experimental study involving traditional solution concepts from the point of view of plasticity and empowerment—for certain kinds of games or dilemmas, what must hold at equilibrium? Can we use the tools of plasticity and empowerment to make predictions about such equilibria? What does cooperation or defection imply about a player’s (or a communities) plasticity and empowerment? Exploring these questions can give further insight into achieving the proper balance.
>
> __5. Safety.__ Lastly, we foresee valuable pathways to connecting to the safety literature. As Turner et al. (2021) explored in Optimal Policies Tend to Seek Power, we speculate that one way to modulate the safety of different systems is through the use of tools like plasticity and empowerment. For example, in a collaborative setting, it would be desirable for agents to act in a way to respect the agency of its collaborators—that is, to ensure that it does not systematically damage the plasticity or empowerment of collaborators. This is again a practical pathway that would enable us to design agents that balance plasticity and empowerment tactfully.
>
> We thank the reviewer for their questions, and are happy to discuss anything further.

---

> > ### Author Response · Authors · 2025-08-05
> > **Follow Up**
> >
> > We believe our rebuttal addresses the reviewer’s four main questions. If Reviewer x8QU feels satisfied with our responses, we kindly request that they reconsider their recommendation to reject the work. Or, if the reviewer has lingering questions, please let us know and we will be happy to discuss further.

---

### Official Review · Reviewer_2PEP · 2025-07-02

**Clarity:** 3
**Significance:** 2
**Originality:** 3
**Rating:** 4
**Confidence:** 3

**Summary:**

This paper develops an information-theoretic mathematical definition for plasticity. Towards this, one key contribution is the formulation of Generalized Directed Information (GDI), which generalizes the notion of directed information between sequence of random variables $I(X_{1:n} → Y_{1:n})$ to handle arbitrary sub-sequences $I(X_{a:b} → Y_{c:d})$

Given this formalism, the paper defines plasticity of an agent w.r.t. an environment as the (generalized) directed information from its observations to actions. Similarly, a definition of the empowerment is formulated as the (generalized) directed information from its actions to observations, and the paper shows that this definition subsumes previous ones that only considered full sequences.

The main results then show that: a) there is a symmetry in the definition of plasticity and empowerment i.e. plasticity = the environment’s empowerment w.r.t. the agent, and b) the sum of an agent’s plasticity and empowerment is bounded i.e. given a specific environment, there is a tradeoff between how plastic an agent can be vs how empowered it is.

**Questions:**

- Some more discussion on the relation of the current definition to the typical semantic of 'neural plasticty' would help.

- I'd also appreciate a discussion/rebuttal of the weakness 2 i.e. it seems that an agent **can** be simultaneously empowered and plastic where its current observations can influence future actions while its current actions can also influence future observations, and there seems to be no tradeoff between these.

**Ethical Concerns:**

["NO or VERY MINOR ethics concerns only"]

**Final Justification:**

After the rebuttal and followup discussion, I am convinced the work is novel and general (specially the GDI and the subsequent definitions of plasticity) as well as technically correct and clear. Personally, I am still not sure about the potential impact and the usefulness of the central results in Theorem 4.8, but I believe these things are better left for the community to determine in time. Overall, I'd be happy to update the score to lean towards acceptance and would encourage the authors to update the text to reflect the  limitations discussed.

**Limitations:**

Yes

**Quality:**

2

**Strengths And Weaknesses:**

Strengths:

- The formulation of GDI is an interesting contribution. It helps generalize the notion of directed information to consider arbitrary subsets and could be applicable beyond the context of plasticity/empowerment. Similarly, the generalized definition of empowerment is also a useful extension of previous ones as it now lets us consider the empowerment in terms of influence of a specific subsequence actions on a subsequence of future observations.
- The main results which highlight that it isn’t possibly to simultaneously optimize empowerment and plasticity is a thought-provoking observation (although some concerns on this below)
- The paper is generally well-written with appropriate levels of background and detail in the main text and appendix. Given the topic and the contribution, this was likely not an easy thing to accomplish!

Weaknesses:

- A primary concern about this work is that it seems to be redefining an existing and widely used term “plasticity” to be rather different from the (intuitive but not mathematically precise) typical definition of “an agent’s capacity to remain adaptive”. In this work, the term “plasticity” is instead better described as capturing “the influence of an environment on the agent” and it’s not clear why we should be using the same word ‘plasticity’ for this measure. As an illustration, in the mouse example in the appendix, the fully stochastic room certainly reduces the empowerment of the agent but it’s not clear there is any useful information to learn for helping future adaptation in the stochastic one. Concretely, let’s imagine a neural network trying to learn to predict the outcome of the agent’s actions. One environment is deterministic, allowing the network to model laws of physics and the other environment is random, and makes the neural network learn to predict a uniform distribution over states. As the typical definition of ‘plasticity’ is how adaptive this network can be to future changes, it’s not clear why the network being learned in the fully stochastic setting will be better suited to adapt to new distributions than the one in the deterministic setting. So although this stochastic environment does maximize ‘plasticity’ as defined in this paper, it does not intuitively do so for the ‘classical’ notion of plasticity.
- While the bound in Egn 19 is interesting, I’m not sure if it’s a very profound result. For example, if there is no overlap in a:b and c:d, one of the terms is trivially zero (as directed influence of future on past will be zero). On the other hand, if the two subsequences overlap fully (e.g. a=b=1, c=d=n), it is essentially the result in Eqn 4. It would have been a much more surprising result had the bound in Eqn 19 been in the direction a:b → c:d for both terms i.e. for non-overlapping intervals, one term were not trivially zero. In the current form, an agent **can** be simultaneously empowered and plastic in terms of its actions affecting future observations, and observations affecting future actions. Specifically, considering non-overlapping intervals a:b (’current’) and c:d (’future’), there seems to be no tradeoff between the current actions’ influence on future observations and current observations’ influence on future actions.

---

> ### Author Rebuttal · Authors · 2025-07-30
>
> __Overview.__ We would like to thank the reviewer for their time in reading and reviewing our paper. We address the reviewer’s two primary concerns below in sections Comment 1 and Comment 2, and will update the paper in light of their comments.
>
> &nbsp;
> # Comment 1: Plasticity Semantics and Example
>
> We break our response to Comment 1 into two parts, focusing on (1a) plasticity semantics and (1b) the rooms example.
>
> &nbsp;
>  ## (1a) Semantics of Plasticity
>
> > “A primary concern about this work is that it seems to be redefining an existing and widely used term “plasticity” to be rather different from the (intuitive but not mathematically precise) typical definition of “an agent’s capacity to remain adaptive”... Some more discussion on the relation of the current definition to the typical semantic of 'neural plasticty' would help.”
>
> Let us consider the following wording for the intuition of plasticity and our mathematical definition:
>
> > (Plasticity intuition) “an agent’s capacity to remain adaptive [to its observations]”.
> >
> > (Our definition) “an agent’s capacity to be influenced by its observations”.
>
> The difference separating these two is very subtle, and is rooted in how one chooses to define adaptivity mathematically. Our formalism gives a mathematical definition of “adaptivity” in terms of the degree to which observations can influence action (as measured by generalized directed information)---this is in line with classical definitions of plasticity and learning as “the ability to change action due to experience” (paraphrasing from Chapter 3 of West-Eberhard, 2003; Lachman, 1997; among others). Consequently, we believe our definition of plasticity does capture the intuitive use of plasticity, and hope that this wording and commentary makes this more apparent.
> &nbsp;
>  ## (1b) Rooms Example
>
> Regarding the example, you mention:
>
> > “...it’s not clear there is any useful information to learn for helping future adaptation in the stochastic one”...
>
> Notice that information is so far neutral in our theory—we focus on the capacity to influence and be influenced, and all information is created equal. For this reason, it is perhaps easy to see why the stochastic case provides more opportunities to respond in your neural network example: there is just a richer set of possible experiences the agent can adapt to.
>
> To study whether it is _useful_ to be plastic or be empowered, we need to introduce a goal for the agent to pursue or a target to predict (as in your example): Then, information is useful if it enables an agent to effectively pursue its goal or predict the target. This is an important area for future work, and we do have some preliminary results in this direction: for any realizable value of plasticity or empowerment, there will exist environment-goal pairs for which all optimal agents have at least the given level of plasticity (or empowerment). So, it is sometimes necessary to be plastic or empowered to be optimal. There are thus paths that connect usefulness to plasticity and empowerment, and we agree that a deep analysis of this relationship is a valuable direction for future work.
>
> > “As the typical definition of ‘plasticity’ is how adaptive this network can be to future changes, it’s not clear why the network being learned in the fully stochastic setting will be better suited to adapt to new distributions than the one in the deterministic setting.”
>
> We again emphasize that it is best to untether plasticity from usefulness, first (we can then incorporate what counts as “useful” information later, as suggested above). In the absence of specific information being useful, we see that we lose the sense of one setting being “better suited” to adaptation—there is only influence, not a notion of “better”. And, the stochastic setting comes with a richer space of experiences for the agent to respond to, so the potential for plasticity is higher.
>
>
>
> References:
>
> * West-Eberhard, M. J. (2003). Developmental plasticity and evolution. Chapter 3. (pp. 34). Oxford University Press.
> * Lachman, S. J. (1997). Learning is a process: Toward an improved definition of learning. The Journal of Psychology.
>
> &nbsp;
> # Comment 2: No Tradeoff for Some Interval Configurations
>
> Our response is again in two parts: (2a) the trade off for other interval configurations and (2b) the value of Theorem 4.8.
>
> &nbsp;
> ## (2a) No Tradeoff for Some Interval Configurations
>
> > I'd also appreciate a discussion/rebuttal of the weakness 2 i.e. it seems that an agent can be simultaneously empowered and plastic where its current observations can influence future actions while its current actions can also influence future observations, and there seems to be no tradeoff between these.
>
> The reviewer is correct in thinking that Theorem 4.8 will not extend to all possible configurations of indices. This is a valid point, and we are happy to add some text to the paper clarifying this. We roughly formalise this as an existence claim that $E + P$ can be greater than $m$ from Theorem 4.8 if we swap the index order, as follows.
>
> __Conjecture 1.__ For all intervals $[a:b]_n, [c:d]_n$ where $b < c$ and $b-a = d-c$ (for simplicity), there exists a pair $(agent, env)$ such that $E_{a:b,c:d}(agent, env) + P_{a:b,c:d}(agent, env) > m$, where $m$ is set as in Theorem 4.8.
>
> _Proof Sketch._ We prove the result by constructing an example that satisfies the inequality for fixed but arbitrary intervals $[a:b]_n$ and $[c:d]_n$ where $b < c$ and $b-a = d-c$. For simplicity let both $\mathcal{A} = \\{0,1\\}, \mathcal{O} = \\{0,1\\}$.
>
> For timesteps $a:b$, consider an agent-environment pair where actions and observations are each samples from a uniform distribution over $\\{0,1\\}$.
>
> Let agent and environment be such that over the $i$th timestep in $c:d$, the observations mirror the $i$th action in $a:b$, and the $i$th action in $c:d$ mirrors the $i$th observation in $a:b$. So, the actions from $c:d$ receive maximal influence from prior observations, and similarly for the observations.
>
> In other words, the empowerment is $\mathbb{H}(A_{c:d})$, which is $k \cdot \log_2 |\mathcal{A}| = k\cdot \log_2(2) = k$. The same holds of plasticity, and therefore we find $E_{a:b,c:d}(agent, env) + P_{a:b,c:d}(agent, env) = 2k$. This is strictly larger than $m$ from Theorem 4.8, and we conclude.
>
> _End Proof_
>
> We will add text to the discussion of Theorem 4.8 to emphasize that it does not apply to the cases involving swapped intervals like the above, and will include a polished version of this conjecture and proof somewhere in the paper or appendix to add clarity.
>
> &nbsp;
> ## (1b) Theorem 4.8 Remains Useful
>
> However, we also want to emphasize that Conjecture 1 and Theorem 4.8 holding simultaneously does not detract from Theorem 4.8, for a few reasons:
> 1. First, for any non-overlapping intervals $[a:b]_n, [c:d]_n$ that satisfy Conjecture 1, there exists a larger pair of intervals $[w:x]_n, [y:z]_n$ that (i) contain the former, and (ii) do overlap, so the tension comes into play and Theorem 4.8 applies. In this way, the tension can effectively always rear its head, though its salience naturally depends on specific aspects of the agent, environment, and the intervals.
> 2. Second, consider sequential intervals of length-t episodes: Theorem 4.8 tells us there must again be a bound on the agent’s plasticity and empowerment within each episode. As a consequence, _accumulating over episodes_, plasticity and empowerment are also bounded. So, an agent must balance between the two quantities over the course of its lifetime in a particular sense. A similar conclusion can be reached following typical arguments about entropy rates: the total rate of information exchanged is upper bounded for finite alphabets A and O. As such, the limiting rate of plasticity and empowerment is also likely to be upper bounded in a way that suggests a lifetime-level tension. Exploring this zoomed-out view from either of the two perspectives mentioned is another rich direction for future work that requires careful analysis, but again we believe the tension highlighted by Theorem 4.8 has potential for far reach.
>
> Lastly, let us emphasize the generality of Theorem 4.8: it holds for effectively all agents and all environments—we make no assumptions on agent nor environment apart from the finiteness of the sets $\mathcal{A}$ and $\mathcal{O}$—the environment can be partially observable, non-stationary, and unbounded. These are very mild assumptions for a result of this kind. The result provides the first connection between the concepts of plasticity and empowerment in a general setting that yields a fundamentally new way to look at agents, and we believe this makes it a valuable contribution in its own right. Lastly, its proof requires more than the standard conservation law of DI (Prop 2.3), but also requires the introduction of the GDI and the expanded conservation law (Theorem 3.5), whose definition and analysis are non-trivial in their own right.
>
> &nbsp;
> # Summary
>
> We will add expanded discussion to the paper that incorporates and clarifies the discussion surrounding Comments 1 and 2. We agree this will bring new clarity to the paper, and appreciate the reviewer’s time and comments.

---

> > ### Author Response · Authors · 2025-08-05
> > **Follow Up**
> >
> > We believe the comments in our rebuttal address the reviewer’s two main concerns.
> >
> > If Reviewer 2PEP feels satisfied with our responses, we kindly request that they reconsider their recommendation to reject the work. Or, if the reviewer has lingering questions, please let us know and we will be happy to discuss further.

---

> > ### Comment · Reviewer_2PEP · 2025-08-06
> >
> > I'd like to thank the authors for the detailed response and the discussions. I would not  argue against the paper as I do feel the formalization of GDI is interesting in its own right and the definitions here are at the very least an interesting contribution. However, there are still some concerns which prevent me from actively recommending acceptance, and I’d be curious to hear the authors’  thoughts on these.
> >
> >
> > ## Plasticity Semantics
> >
> > Thanks a lot for the discussion on disentangling plasticity from utility. I think that is indeed a helpful perspective and I found the description stating "information is so far neutral in our theory—we focus on the capacity to influence and be influenced" to be a really great and succinct summarization.
> >
> > However, I am worried that this makes the current framework a bit difficult to apply in practical scenarios where there are notions of utility. Considering again the stochastic vs deterministic room example for an agent, there is certainly more information in the former (and thus a higher plasticity), but I am struggling to think of a concrete practical case where this implies better adaptability for the learned agent e.g. what examples of utility would correspond to the agent being able to learn better from the first environment so that it aids downstream adaptation?
> >
> >
> > ## Tradeoff
> >
> > Thanks for the confirmation that the tradeoff is not there for the reversed order. However,  I don't completely follow the arguments in (2b) above highlighting the utility of the main theorem, but let me preface that by discussing a couple cases.
> >
> > Let us consider two extremes: I) non-overlapping intervals, and II) completely overlapping intervals. In scenario I, let us further consider Ia) intervals as in 4.8, and Ib) reversed intervals. For Ia, we know the bound is trivial because future can't influence the past so the theorem doesn't tell us much. For Ib, as discussed in Conjecture 1, the bound does not apply. For scenario II, the bound would just follow from Eq 4 (as the sum of directed mutual information in both directions would be bounded by information in one of the variables).
> >
> > Now back to the points in 2b). The first point states that “tension comes into play” where there is overlap. So let us examine the case with maximum ‘tension’ i.e. fully overlapping intervals. As discussed above, the bound in this directly comes from Eq 4 — but is the claim that this bound is indeed surprising/revelatory (limiting to fully overlapping intervals)? Basically, I am having a difficult time understanding the importance of the central result because:
> >
> > - Considering the extreme of non-overlapping intervals, the tradeoff is either trivial (as future can’t influence past so one term is always zero) or does not apply (for reversed intervals)
> > - Considering the other extreme of fully overlapping intervals, the derived bound does not seem very profound as well
> > - Given that there are just 3 cases (non-overlapping, fully overlapping, somewhat overlapping) one of the above must be true: a) the bound for one of the two extremes is indeed revelatory, in which case I’d like to better understand why that is, or b) even if the main result does not reveal something fundamental about the two extremes, the result for somewhat-but-not-fully-overlapping intervals reveals something fundamental — if so, I’d like an intuitive explanation of what this is and why it’s not apparent in either extremes?
> >
> > Lastly, for the case of sequential intervals of length-t, while there is a bound on the plasticity and empowerment within each episode, there isn't a corresponding tight bound on plasticity and empowerment across the sum of sequences as a whole (as what the agent observes in the first sequence can affect its actions in the future sequence while also  and there is no trade-off here as discussed in Conjecture 1).

---

> > > ### Author Response · Authors · 2025-08-06
> > > **Response to Official Comment by Reviewer 2PEP**
> > >
> > > &nbsp;
> > >
> > > We thank the reviewer for their continued thoughtful engagement and questions
> > >
> > > &nbsp;
> > >
> > > ## (1) Influence and Utility
> > >
> > > > “... I am worried that this makes the current framework a bit difficult to apply in practical scenarios where there are notions of utility.”
> > >
> > > By developing our theory around influence, we avoid prematurely tying plasticity to specific notions of "useful" information, which can vary greatly depending on how we conceive of goals. We thus maintain the generality of the theory by remaining agnostic to any specific account of goals. Then, we can integrate different conceptions of goals (such as utility), and the theory can accommodate these choices with ease.
> > >
> > > For instance, we have proven results that incorporate reward maximization into the theory. This is done without issue: reward is an element of each observation, and everything in the theory applies. The same can be done with classification, MARL, and so on.
> > > ### (1a) Example
> > >
> > > > “...what examples of utility would correspond to the agent…”
> > >
> > > We design an example to highlight the following intuition: if an agent is not influenced by the reward it observes, then how can the agent learn to maximize reward?
> > >
> > > __Example: Switching Bandit.__ Consider a Bernoulli bandit environment, with each arm parameter theta set to 0 or 1. When theta=0, the reward emitted is 0, and when theta=1, reward is 1 (and observations=rewards).
> > >
> > > Now, suppose that every 50 timesteps the environment randomizes each arm’s parameter over \\{0,1\\}. Each agent’s goal is to maximize reward over its lifetime (say, some fixed T >> 50).
> > >
> > > Notice that all optimal agents have non-zero plasticity. Each agent that continually finds +1 arms will be influenced by what it has observed: recent past rewards.
> > >
> > > It is in this sense that plasticity-as-influence can accommodate utility: plasticity is necessary to learn to pursue goals in uncertain worlds. In cases where the agent’s goal (such as its reward) depends on latent variables of the environment, learning about those latent variables is important. Plasticity means agents are receptive to being influenced by information about those latent variables.
> > >
> > > We emphasize the example is simple to isolate the relationship between plasticity and utility, but note this general story extends far beyond this case.
> > >
> > > Lastly, we point out that Empowerment was also conceived of as influence (see original by Klyubin et al.), and later work has bridged the connection between empowerment and utility (Leibfried et al. 2019, among others).
> > >
> > > &nbsp;
> > > ## (2) Tradeoff, Value of Theorem 4.8
> > >
> > > We believe Theorem 4.8 highlights a new, compelling tension. Naturally there exist cases where the tension does not apply—similarly, the explore-exploit dilemma is important, but of course does not apply to every setting such as standard supervised learning (e.g. PAC). Explore-exploit is not uninteresting because there exist cases where it does not apply, it is interesting because there are cases where it does.
> > >
> > > We suggest the plasticity-empowerment tension is similar: the fact that there are cases where this tension occurs is interesting, and makes it valuable and worth researching further.
> > >
> > > > “Given that there are just 3 cases (non-overlapping, fully overlapping, somewhat overlapping) one of the above must be true…”
> > >
> > > We agree: the case of no overlap does not pose a meaningful tension.
> > >
> > > The other two cases (which are the significant ones) remain interesting to us. Our understanding is that the reviewer feels the full overlap case is uninteresting because it follows directly from Eq 4, so we focus our answer there, first. We have two comments:
> > > 1. First, we respectfully disagree. The proof of Theorem 4.8 is not a direct consequence of Eq 4, even in the full overlap case (a=c, b=d). The result only follows from Eq 4 in one specific case in which the GDI collapses to DI (a=c=1, b=d=n). Our result is more general (every [a:b]_n, [c:d]_n, with the tension manifesting in interval pairs with overlap), and this general form requires the careful design of the GDI and its analysis (Props 3.2-3.4, C.1-C.3, Theorem C.1), and the new conservation law (Theorem 3.5). Unlocking this definition, clarity, and generality was non-trivial.
> > > 2. Second, we do not believe that proof complexity is a necessary property of compelling theoretical work: identifying novel connections (such that the proof may be obvious in hindsight) is valuable. Many results from AI and ML rehash tools (Jensen’s/Hoeffding’s inequalities, union bounds,, and so on). It can be valuable to uncover novel connections and insights regardless of proof complexity. That being said, we maintain the proof is non-trivial in light of the points made in (1.)
> > >
> > > We also maintain that the directions regarding a lifetime-level tension remain compelling (both via the episodic view or limiting view), and make for compelling future work.
> > >
> > > &nbsp;
> > >
> > > We thank the reviewer again for their time and questions, and we are happy to discuss further.

---

> > > > ### Author Response · Authors · 2025-08-08
> > > > **Final Thoughts**
> > > >
> > > > Dear Reviewer 2PEP,
> > > >
> > > > We recognize the time remaining in the discussion period is short, with <24 hours remaining: we believe we have resolved your concerns about the work. If you do have any final reactions to our above points, we are happy to discuss further. We will do our best to turn around any final comments responding before the end of the discussion period.
> > > >
> > > > To summarize, we believe the work:
> > > > 1. __Is Novel__: We are the first to present a general mathematical definition of plasticity, and it sounds as though following our discussion, the reviewer agrees our definition does in fact reflect the intuitive use of the term (as do other reviewers). We are also the first to develop and analyze the generalized directed information, which also allows us to strictly generalize existing definitions of empowerment. Lastly, we are the first to discover and elucidate the relationship between plasticity and empowerment. We believe these all represent conceptually rich and technically rigorous novelties of the work.
> > > > 2. __Is General__: The main results and insights come about under a highly general theoretical frame---the assumption we make is that the action and observation spaces are finite. The environment and agent can otherwise be arbitrary. This is a very mild assumption for work of this kind, and can elevate the potential reach of the definitions, results, and perspectives.
> > > > 3. __Is Correct__: It seems we are all in agreement that the results are correct (the proofs are valid).
> > > > 4. __Is Clear__: We believe that despite the complexity of the topics involved, the paper clearly articulates these ideas and findings (as the reviewers have pointed out).
> > > > 5. __Has High Potential For Opening Pathways__: Lastly, we maintain that in part due to points 1-4, there is a high potential here to open new research pathways across different fields. For instance, in our rebuttal-discussion together, we have been able to pose new questions about plasticity, utility, and empowerment; this is a good sign about the power of the theory to make new research questions actionable---we suggest that the ability to open new actionable research questions is a considerable strength for new theory, and should be taken as a strength of this work.
> > > >
> > > > It sounds as though the reviewer's main critique is that Theorem 4.8 is potentially uninteresting to them (though as argued above, we still believe Theorem 4.8 is interesting: we are happy to discuss further).
> > > >
> > > > In light of the strengths of the work enumerated above, and absent of any glaring weakness (we believe we addressed the point regarding the semantics of the term, which has been backed up by the other reviewers), we kindly request that Reviewer 2PEP reconsider their score.
> > > >
> > > > We are happy to discuss further should the reviewer feel anything has not been touched upon.
> > > >
> > > > Thank you for your consideration

---

> > > > > ### Comment · Reviewer_2PEP · 2025-08-08
> > > > >
> > > > > Thank you for the followup and the summary. I agree with the comments 1-4 above in that the work is novel and general (specially the GDI and the subsequent definitions of plasticity) as well as technically correct and clear. Personally, I am still not sure about the "high potential" and the usefulness of the results in Theorem 4.8, but I believe these things are better left for the community to determine in time instead of the subjective choices of a specific reviewers. Overall, I'd be happy to update the score to lean towards acceptance.

---

### Decision · Program_Chairs · 2025-09-17

**Decision:**

Accept (spotlight)

**Comment:**

This paper develops an information-theoretic framework for understanding agent-environment interactions by introducing a mathematical definition of "plasticity" - the extent to which an environment can influence an agent's behavior. This uses a novel concept called Generalized Directed Information (GDI) that extends traditional directed information to handle sequences of different lengths. The key insight is that plasticity and empowerment (an agent's ability to influence its environment) are mirror quantities that exist in a fundamental trade-off relationship, meaning that, in general,  both cannot be simultaneously maximized. The work demonstrates that agents and environments are symmetric communication partners, with the agent's plasticity corresponding to the environment's empowerment, and establishes theoretical bounds showing these desirable qualities are inherently in tension with each other.

Reviewers identified the following strengths: The formulation of Generalized Directed Information (GDI) represents an interesting theoretical contribution that extends directed information to arbitrary subsets and has potential applications beyond plasticity/empowerment. The main theoretical results demonstrate the impossibility of simultaneously optimizing empowerment and plasticity provide thought-provoking insights. For instance, in multi-agent environments, each agent belongs to the environment of other agents, and for things such as cooperation to emerge, individual agents must accept limitations on their empowerment. The paper is generally well-written with appropriate background detail.

More significantly, the majority of reviewers praised the work's conceptual clarity and broad relevance, noting that it proposes a crisp, general notion of plasticity requiring minimal assumptions about agent-environment interaction, with the mirror relationship between plasticity and empowerment being conceptually appealing and well-integrated with existing literature. The theoretical framework raises compelling questions about optimal agent design and the balance between plasticity and empowerment. We might expect wide appeal and lasting impact across diverse areas from reinforcement learning theory to modern language model agents and recommender systems.

In terms of weaknesses, there was discussion between the authors and some reviewers about the choice of the term "plasticity", but the authors give a solid defence of this choice. There are also some technical discussions regarding the meaningfulness of the result and possibility of actionable insights. However, the authors rebut these effectively in the first instance, and it may be that the community response to this paper will be needed to determine the full scope of this. Finally, there was a request for experimental evaluation of the ideas in the paper.

Based on the paper, reviews, rebuttals and responses, I believe the paper to be of high quality and worthy of acceptance at conference.